# Updated monthly and new daily bias correction for assimilation-based passive microwave SWE retrieval

Pinja Venäläinen[1], Colleen Mortimer[2], Kari Luojus[1], Lawrence Mudryk[2], Matias Takala[1], and Jouni Pulliainen[1].

[1]Finnish Meteorological Institute, PO Box 503, FIN-00101 Helsinki, Finland.

[2]Climate Research Division, Environment Climate Change Canada, Toronto, Canada

*Correspondence to*: Pinja Venäläinen (pinja.venalainen@fmi.fi)

**Abstract**. Snow water equivalent (SWE) is a valuable characteristic of global snowpack, and it can be estimated using passive spaceborne radiometer measurements. The radiometer-based GlobSnow SWE retrieval methodology, which assimilates weather station snow depth observations with passive microwave brightness temperatures, has improved the reliability and accuracy of SWE retrieval when compared to stand-alone radiometer passive microwave (PMW) methods. However, even this assimilation-based method fails to estimate large (> 150 mm) SWE values as snow changes from a scatterer to an emitter. Correcting for these systematic biases can improve PMW-based SWE estimates, especially for high SWE magnitudes. Previously, a monthly bias correction using snow course observations was applied to the GlobSnow v3 product for February – May. This method reduced the spread in March SWE estimated from four gridded products. In this research, we use newly available snow course data to update this bias correction and expand it to cover the months of December through May; we also extend the monthly bias correction to a daily bias correction. The new monthly and daily bias corrections are applied to an updated version of the GlobSnow product - Snow CCI v3.1 product. The Northern Hemisphere climatological snow mass from the Snow CCI v3.1 bias corrected products (daily and monthly) is consistent with that from a suite of reanalysis products. This represents a significant improvement for the months of April and May compared to the original GlobSnow v3.0 bias corrected product, as is the provision of daily bias corrected SWE estimates.

## 1 Introduction

Snow water equivalent (SWE), defined as the depth of water that would result if the snowpack were to melt completely, plays a pivotal role in water resource management, climate modelling, flood prediction, and ecological studies (Hall et all, 2008; Magnusson et al., 2020; Derksen and Brown, 2012; Jones et al., 2011). Passive microwave (PMW) radiometer observations, which provide near-continuous brightness temperature (Tb) measurements dating back to 1978, can be used to estimate SWE. PMW SWE retrieval methods rely on the brightness temperature difference between two channels. Tb measurements at a frequency insensitive to dry snow (around 19 GHz) serve as a baseline, which are compared with Tb measurements at a frequency sensitive to dry snow (around 37 GHz). The latter wavelength is closer in scale  to the snowpack microstructure, which induces significant volume scattering and

attenuates signal (Chang et al., 1987; Kelly et al., 2003; Mätzler, 1994). Significant uncertainties limit SWE retrievals based solely on radiometer measurements, and their accuracy often fails to meet user accuracy requirements in terms of retrieval skill (e.g. Derksen et al. 2022; GCOS 2022) and exhibit poor spatial and temporal correlation with other SWE products (such as the NASA Global Land Data Assimilation System version 2 – GLDAS-2; the European Centre for Medium-Range Weather Forecasts (ECMWF) interim land surface reanalysis – ERA-Interim/Land and ECMWF Reanalysis version 5 – ERA5 and the Crocus snow model driven by ERA-Interim meteorology) (Derksen et al., 2005; Mudryk et al., 2015; and Mortimer et al., 2020).

Assimilation of in situ snow depth observation can improve the accuracy of PMW-based SWE retrievals (Pulliainen et al. 2006). This assimilation approach was used in the European Space Agency (ESA) GlobSnow project, and its development continues in the ESA Snow CCI+ project. Despite improvements under the Snow CCI+ program (Mortimer et al. 2022), the method is still limited by the inability of passive microwave observations to estimate large SWE values as the snowpack changes from a scattering medium to a source of emission when the snowpack is deep (SWE ~ > 150 mm). This occurs because, at higher frequencies (~37 GHz), snowpack transitions from a scattering medium to an emitter when SWE exceeds ~150 mm, reducing sensitivity to further SWE increases.One approach to overcome this limitation is to apply a bias correction. Pulliainen et al. 2020 demonstrated that the magnitude of the bias in SWE estimates from GlobSnow 3.0 (GSv3.0) relative to in situ snow course observations is stable through time but exhibits a strong spatial pattern. Correcting for this spatial bias can, therefore, improve the estimation of hemispheric-scale snow mass. Pulliainen et al. (2020) applied this concept to four snow products: MERRA2, GlobSnow GSv3.0, and the Crocus and Brown snow models, both of which were forced by ERA-Interim. This reduced the spread in March SWE estimates from 33 % to 7.4 %.

Although this method has been used to produce monthly bias corrected GlobSnow v3.0 products for February through May, only the March SWE time series has been thoroughly evaluated (Pulliainen et al. 2020, Luojus et al. 2021). March has been the focus of the evaluations as snow mass usually peaks during this month. Furthermore, until now, insufficient snow course data precluded bias correction outside these months (Luojus et al. 2021). Given the demonstrated success of this method, in this study, we apply the method to the most recent product in the GS/CCI product line – Snow CCI v3.1 (SCv3.1). We exploit the availability of additional snow course data, which has been made available since GSv3.0, to improve the bias correction and extend it to December and January. Building on Pulliainen et al. (2020) and Luojus et al. (2021), which limited the evaluation of bias-corrected products to March, we analyse the bias corrected SWE estimates for all months from December to May. Finally, to address user needs (Derksen et al. 2022, GCOS 2022), we developed a daily bias corrected SCv3.1 product that is based on monthly bias correction fields.

## 2 Data and Methods

### 2.1 SWE retrieval

The PMW SWE retrieval is based on the methodology introduced by Pulliainen (2006) and Takala et al. (2011) and is briefly summarised here. The two primary data inputs to the algorithm are vertical passive microwave Tb and daily synoptic snow depth (SD) measurements. SD measurements are collected from multiple sources. The main sources for

Eurasia are the European Centre for Medium-Range Weather Forecasts (ECMWF) and the All-Russia Research Institute of Hydrometeorological Information - World Data Cente (RIHMI-WDC) (Bulygina and Razuvaev, 2012).

Global Historical Climatology Network daily (GHCNd) (Menne et al., 2012) by National Oceanic and Atmospheric Administration (NOAA) is used as the main dataset for North America. The satellite Tb data are from the Special Sensor Microwave/Imager (SSM/I) and Special Sensor Microwave Imager/Sounder (SSMIS) instruments on board the Defence Meteorological Satellite Program (DMSP) F-series satellites. Measurements at 37 GHz and 19.40 GHz (SSM/I) or 19.35 GHz (SSMIS) are used for SWE retrieval. Both synoptic SD and Tb measurements are filtered before

the algorithm ingests the data. Filtering is needed to guarantee convergence on a solution during the assimilation process, and the filtering process is described in detail in Luojus et al. (2021). Water, complex terrain, and dry snow masking are applied to Tb measurements. SWE retrieval is performed only for dry snow; for wet snow, the SWE estimates are based on the background SD field.

The four main steps of the SWE retrieval are described shortly here; for more details, see Luojus et al. (2021). Firstly,

kriging interpolation is used to produce a continuous field of in situ SD and its variance using filtered synoptic SD observations for the day under investigation. Then, the effective snow grain size (diameter), $d_0$, is retrieved for grid cells with SD observations (measurements, not interpolated values) by numerical inversion of the multi-layer HUT (Helsinki University of Technology) (Pulliainen et al., 1999) snow emission model. The model is fitted to PMW Tb observations at the locations of SD observations by optimizing the value of $d_0$. The final $d_0$ estimate and its standard

deviation at each SD measurement location is obtained by calculating the average value of the six nearest SD measurements.

Thirdly, a background $d_0$ field (and its variances) is interpolated from the $d_0$ estimates produced for pixels with SD observations in the previous step. Finally, SWE is retrieved by ingesting observed Tb, retrieved effective snow grain sizes, and grain size variances into a numerical inversion of the HUT snow emission model. The HUT model estimates

are matched to observations numerically by incrementing the SD value. The background SD field (produced in the first step) is used to constrain the retrieval. The assimilation procedure adaptively weighs the Tb measurements and the background SD field to produce a final SD estimate, which is converted to SWE using the constant snow density (value of 240 kg m$^{-3}$ is used for snow density, as this is a reasonable global value given by the analysis of Sturm et al. (2010)) and a measure of the statistical uncertainty (variance estimate) for each pixel. After these four main steps are performed,

snow-free areas are identified using various snow masks and cleared of SWE to form final SWE estimate maps.

## 2.2 Snow CCI v3.1 CDR

Although the general framework has remained consistent in subsequent versions of the GlobSnow and Snow CCI SWE products, modifications have been made to the retrieval algorithm and the input data, that have improved the accuracy

of the SWE retrieval. Here we outline key differences between the SCv3.1 climate data record  and the older GSv3.0 dataset to which the previous bias correction was applied. First, SCv3.1 uses the NASA MEaSUREs Calibrated Enhanced-Resolution Passive Microwave Daily EASE-Grid 2.0 Brightness Temperature ESDR (Brodzik et al., 2016) instead of the heritage Nimbus-7 (1979-1987) (Knowles et al., 2000) and SMMR (1988–present) (Armstrong et al., 1994) Pathfinder Daily EASE-Grid 1.0 Brightness Temperature datasets. The newer recalibrated enhanced resolution

PMW data allowed SCv3.1 to be generated at a finer spatial resolution (EASE-Grid 2.0 12.5 km re-gridded to 0.1°

lat/lon) compared to GSv3.0 (EASE-Grid 1.0 25km) and improved the continuity of the SSM/I – SSMIS time series (Mortimer et al. 2022).

Second, SCv3.1 utilises spatially and temporally varying snow densities in the retrieval instead of the constant density (240 kg m$^{-3}$) used in GSv3.0. Snow density serves as one of the inputs to the HUT snow model employed in the retrieval, aiding in determining effective snow grain sizes and SWE. This change in snow density parameterization improved the overestimations of small SWE values and brought the timing of peak SWE closer to that of other gridded SWE products (Venäläinen et al., 2023).

Third, the dry snow detection algorithm used in the retrieval has been updated, and the snow masks used to remove SWE estimates from snow-free areas during post-production have both been updated. Inside the retrieval, dry snow is detected using a modified version of the Hall et al. (2002) algorithm, which was applied in GCv3.0. The updated algorithm has different threshold values than the original algorithm, and this improves dry snow detection, especially during snow accumulation season when the original algorithm often under-detects snow (Zschenderlein et al., 2023). The threshold for SD was decreased from 80 mm to 30 mm, the brightness temperature thresholds were changed from 250 K to 255 K for Tb37V and from 240 K to 250 K for Tb37H. In post-production, SWE estimates are removed from snow-free areas using a combination of optical and passive microwave snow extent information. GSv3.0 used a passive microwave thresholding approach by Takala et al. (2009) and the JASMES 5 km Snow Extent data product (Hori et al., 2017). The SCv3.1 product replaces the JASMES 5km SE data with CryoClim snow cover extent (Solberg et al., 2014), supplemented with data from the passive microwave thresholding approach.

Finally, extending the time series to include more years will impact the filtered SD data. Before performing spatial interpolation and assimilation (Sect. 2.0), the synoptic SD data are filtered to exclude stations with fewer than five years of data and those where the mean SWE exceeds 150 mm for half of the recorded period. Since SCv3.1 includes four more years of data than GSv3.0, this filtering protocol may result in slight differences in the SD data input into the SWE algorithm.

### 2.3 Bias correction

### 2.3.1 Monthly bias correction

Assessing and correcting for biases in SWE products requires in situ SWE observations. Snow courses have traditionally been the preferred type of in situ data to evaluate coarse resolution gridded SWE products because they sample at spatial scales of several hundreds of metres to several kilometres. Unfortunately, snow course observations are infrequent (made every 5 days to just once a month), and their locations are unevenly distributed across the Northern Hemisphere. The bias correction method developed by Pulliainen et al. 2020 and applied here is based on the premise that the bias is stable through time but exhibits a strong spatial pattern. By exploiting this temporal stability, we can minimise the impact of infrequent sampling by pooling the bias at each grid cell over the full observational period. In this way, the method addresses systematic spatial biases, but interannual variability in the time series and its bias is retained.

The monthly bias correction strategy is implemented as follows. A mean SWE $BIAS_i$ (in mm) is calculated relative to the reference observations at snow course $i$ from all observations of that particular snow course over the period of

record. All measurements within the same EASE-Grid cell are considered to be from the same snow course location. The SWE reference observation is denoted as $REF_{i,t}$, for snow course $i$ at time step $t$, and $EST_{i,t}$ is the corresponding passive microwave-based estimate. We can calculate the bias for snow course $i$ across the whole time series by:

$$BIAS_i = \frac{1}{N_i}\sum_{t=1}^{N_i}(EST_{i,t} - REF_{i,t}). \tag{1}$$

After the mean bias is calculated for each grid cell with coincident snow course observations, ordinary kriging interpolation is used to create a spatially continuous bias field.

This process is repeated for each month separately, from December through May. Bias fields are not calculated for other months as limited reference data are available, and the snow cover extent is relatively small. A single bias field (interpolated mean bias from each EASE-Grid cell with snow course observations) is produced for each month. It is applied to all years in a time series of monthly SWE maps for the corresponding month. For GSv3.0, monthly SWE maps are the arithmetic mean of the valid SWE retrievals for each pixel. For SCv3.1, days without valid retrieval are first filled with mean estimates from the two closest available retrievals, and then the pixel-wise monthly mean is calculated. Filling missing days before calculating monthly values has a minimal effect during mid-winter when most days have valid retrievals. In May, filling removes little snow to monthly mean values. Bias fields are computed for all land areas north of 15°N and applied to the snow-covered area.

### 2.3.2 Daily bias correction

To expand the usage of the spatial bias correction methodology, we produced daily bias fields and applied them to the daily SCv3.0 SWE product. The daily bias maps were interpolated from the monthly maps as a weighted mean between the 15th of each month. For example, the bias map for 14 January is the weighted mean of December and January maps, and the map for 16 January is the weighted mean of January and February maps. The bias map for 15 January is the same as the January monthly bias map.

Weights are calculated for each day $i$ as follows:

$$w_{1,i} = \frac{d_b - d_i}{d_b} \tag{2}$$

$$w_{2,i} = \frac{d_i}{d_b} \tag{3}$$

where $w_{1,i}$ is the weight of the bias map of the first month for $i^{th}$ day, $w_{2,i}$ is the weight for the bias correction map of the second month for $i^{th}$ day, $d_b$ is the total number of days between the 15th of the first and second month and $d_i$ is the $i^{th}$ day from the 15th of the first month. One map is made for each day between 1 December and 31 May. Daily values for the first half of December (1-15 December) and the second half of May (16-31 May) are assigned the monthly values. These daily bias maps are used to perform bias correction for all years between 1980 and 2022 by subtracting the bias in each pixel from the estimated SWE value in the corresponding pixel.

### 2.4 Use of in situ snow data within the SWE retrieval

In this paper, we focus on updates to the bias correction. However, to interpret the results, it is instructive to understand how and where in situ snow information is used within the retrieval. In situ SWE and snow density information from a precursor to Mortimer and Vionnet (2024), with additional snow density information over Finland, are used to parameterise snow density in SCv3.1 and to generate bias maps. Although both the density fields and bias correction use snow course data, they include different observations and employ slightly different data aggregation methods

First, not all snow courses report snow density or provide SWE and SD from which bulk snow density can be derived. Thus, there are some snow course locations that are included in the bias correction but are not informative for the density fields. Second, to increase the spatial and temporal coverage of snow density information, automated snow pillows with coincident SD measurements are used. In contrast, SWE information from snow pillows are not used to calculate the spatial bias fields.

Third, data aggregation and interpolation methods vary between density and bias correction. To generate daily snow density fields, all density observations within an EASE grid cell over a moving 10-year window are averaged and spatially interpolated to create a continuous field, yielding a daily density field for each day over the period of record. In contrast, for bias correction, monthly biases between the snow course and SWE estimates are averaged across the entire period, producing a single bias map for each month over the study period.

## 2.5 Summary of changes in the reference snow course data

The availability of snow course data, and, in particular, its spatial distribution, will impact the ability to represent and correct for the spatial bias accurately. Since the development of the GSv3.0 of bias fields, more snow course data have become available. The new bias fields (monthly and daily) are calculated using snow course data from North America (Mortimer and Vionnet, 2024), Finland (received from Finnish Environment Institute, Kuusisto, 1984; Haberkorn, 2019), and Russia (Bulygina et al., 2011). Notably, despite the addition of a considerable amount of new in situ data, the assumption that the monthly bias at a given location (EASE grid containing snow course(s)) is stable through time remains generally valid (Appendix A).

Figure 1 presents the locations of reference SWE sites from December to May, with new locations in red and the original in blue. The updated and original snow course datasets have similar locations in Eurasia, except for a few changes. The updated dataset contains around 100 new locations and about 3 000 more observations for Finland. Additionally, the Russia dataset was changed from INTAS-SCONE (Kitaev et al., 2002) to RIHMI-WDC (Bulygina et al., 2011). The new Russian dataset has about 25 00 more observations than the orginal dataset for the comparable period of February-May 1979-2016.

In North America, the new dataset has expanded the coverage across Alaska and the western and northeastern US. There are also several new sites in the northern boreal forest (Quebec and northern Manitoba). As illustrated in Figure 2, these additional sites have increased the number of SWE observations in all months analysed. The amount of data available for bias correction over North America in the lowest (0-50 mm) and highest (150 mm-350 mm) SWE bins has increased significantly (by a factor of >3 in the low bins, and there was previously minimal data in the highest bins). As will be discussed in Sect. 3.1, this additional data in the high SWE bins is responsible for most of the differences in the

bias fields calculated with the updated and original data. It is notable that, especially in the new reference dataset, the reference SWE covers a much larger range in North America compared to Eurasia, and the mean SWE value is larger.

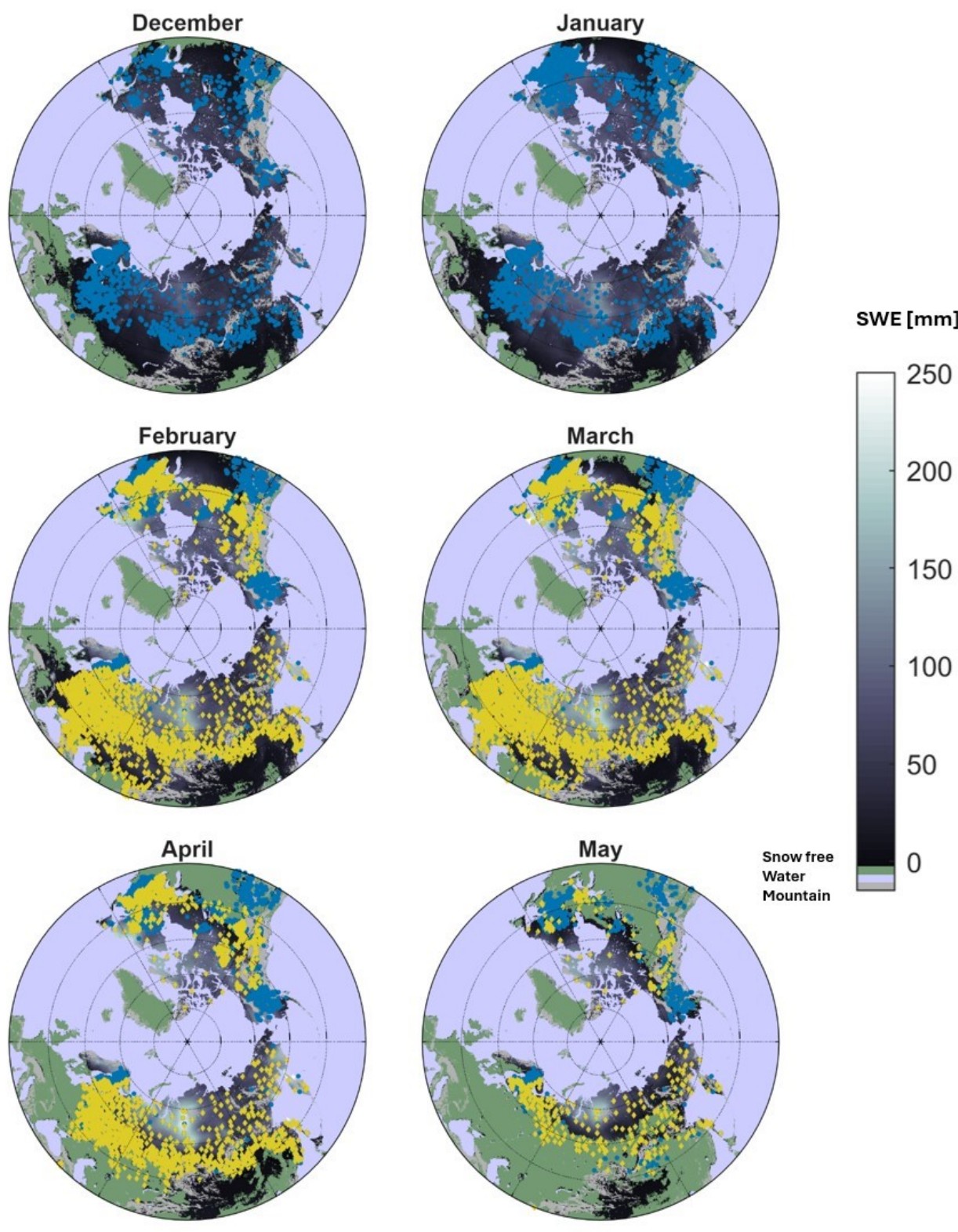

**Figure 1: Locations of reference snow courses. Updates locations are shown in red, and the original ones in blue. Mean monthly SWE and values and snow cover extent are also shown in the figure.**

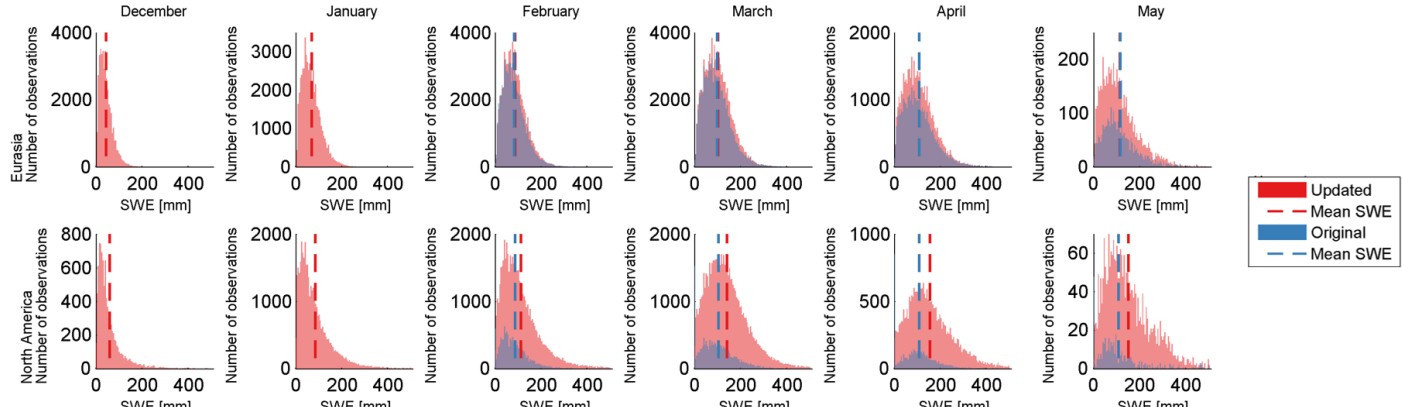

**Figure 2: Monthly distributions of the updated (red) and original (blue) reference SWE measurements for Eurasia (top) and North America (bottom). Monthly mean SWE values [mm] are shown with vertical lines.**

## 2.6 Product comparison and evaluation

First, to isolate the impact of additional snow course sites on the spatial bias field, bias maps for the monthly GSv3.0 product using both the original and updated snow course data are compared. Differences in the bias maps and the corresponding bias-corrected SWE are interpreted in the context of changes to the reference snow course data..

After assessing the impact of additional reference data on the mean monthly spatial bias of the GSv3.0 product, we calculate and apply a monthly bias correction using the updated reference data to the SCv3.1 product. Similar to the previous comparison, we directly compare the bias maps. In this comparison, differences in the bias fields largely reflect changes in the retrieval algorithm and input data, which have been analysed elsewhere (e.g. Mortimer et al. 2022). To understand changes in the final SWE products, we compare the bias corrected SWE of monthly GSv3.0 and SCv3.1 bias corrected products (both using the updated snow course data for bias correction) on a pixel-wise level and investigate their respective time series of March continental (North America and Eurasia) and hemispheric SWE.

**Table 1: The six different SWE products evaluated and analysed in the results section.**

| Product | Time resolution | SWE retrieval algorithm | Bias correction reference data |
|---|---|---|---|
| GSv3.0 | Daily | Original | No bias correction |
| GSv3.0 monthly bias corrected original | Monthly (February-May) | Original | Original |
| GSv3.0 monthly bias corrected updated | Monthly (February-May) | Original | Updated |
| SCv3.1 | Daily | Updated | No bias correction |
| SCv.3.1 monthly bias corrected | Monthly (December-May) | Updated | Updated |
| SCv.3.1 daily bias corrected | Daily | Updated | Updated |

Finally, we validate SCv3.1 daily bias corrected product. Validation of the daily bias-corrected products is challenging because of a lack of independent in situ reference data. Snow course data that would typically be used to validate the SWE products (e.g. Mortimer et al. 2020, 2022; Mudryk et al. 2024) are used to derive spatially and temporally varying snow densities applied in SCv3.1 and to calculate the bias correction fields applied to both GSv3.0 and SCv3.1 (Sect. 2.3).. However, averaging and interpolation steps are applied to these data and automated data are included to compute the density fields. This means that the individual in-situ samples are not fully correlated with the bias-corrected (or non-bias-corrected) SCv3.1 estimates, nor are they fully independentThe impact of the connection between the reference data and the product was demonstrated in the evaluation of monthly GSv3.0 (Luojus et al.2021), where the bias of the uncorrected data was shown to be roughly equal to the bias-corrected data less the value of the correction field at the points sampled. For these reasons, comparison with in situ snow courses, provided in Appendix D, is not a rigorous assessment of product accuracy and thus only serves as a guide to illustrate the impact of bias correction.

Given the lack of independent reference snow courses, we also conduct an evaluation using reference observations from airborne gamma SWE estimates available over the US and parts of southern Canada (Carroll, 2001). Figure 3 shows locations of gamma SWE measurements for March and April. Locations for January and February are similar to March. December, and May contain only a few data points. These data have previously been used to validate gridded SWE products, including GlobSnow (Cho et al. 2020; Mudryk et al. 2024). Airborne gamma observations are typically conducted once per year near peak SWE. Less than one third of site-years have more than one observation. Observations are concentrated in February and March (32% and 38% of observations, respectively). Observations in December, and May each account for less than 1% of the data. Given the limited spatial (and temporal) coverage of these data, the validation with airborne gamma data are not representative of the hemispheric-scale performance but nonetheless provides an important independent baseline. Validation metrics, calculated from coincident reference and product SWE for SWE < 500 mm and SWE < 200 mm, include root mean squared error (RMSE), mean absolute error (MAE), bias, and correlation.

To increase the coverage of our assessment, we include an intercomparison using ensembles of reanalysis products. Pixel-wise comparisons are conducted for the daily CCI bias-corrected product for each month from December to May. NH Hemispheric SWE is compared to two suites of reanalysis products from the SnowPEx Intercomparison Project (Mudryk et al. 2024).

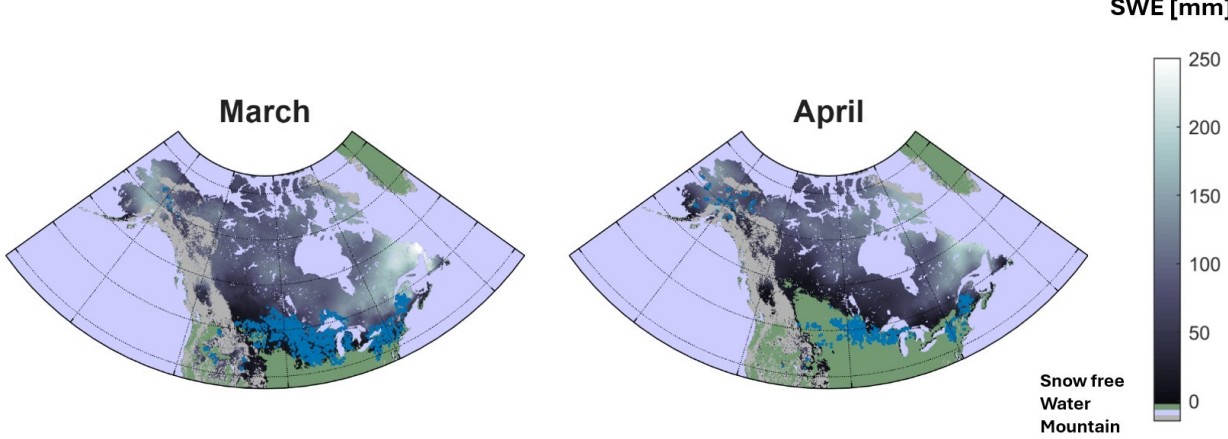

**Figure 3: Distribution of airborne gamma reference data used for validation for March and April. Mean monthly SWE and values and snow cover extent are also shown.**

## 3 Results

### 3.1 Impact of reference data changes on the bias correction

The changes to the reference data described in Sect. 2.3 are expected to impact the accompanying bias fields. For example, a considerable proportion (~ 30 %) of the added data in North America is above the PMW retrieval method detection limit (~150 mm, Luojus et al., 2021), resulting in negative biases. It is expected that the bias over North 280 America will be more negative, and hence, more SWE will be added to the bias corrected product when calculated using the updated reference data compared to the original. In the following, we compare the bias fields for the monthly GSv3.0 dataset using both the updated and original reference snow course data.

Figure 4 shows bias fields for February, March, April, and May calculated using original (top) and updated (middle) reference datasets. Bias fields for the SCv3.1 are also shown in the bottom row of Figure 4. SCv.3.1 bias fields for 285 December and January can be found in Appendix B. The bias fields calculated with the original and updated snow course datasets for GSv3.0 exhibit similar spatial patterns. Both fields have notable negative biases in western North America and the province of Quebec, Canada, for all months, consistent with patterns documented elsewhere (Luojus et al. 2021, Mudryk et al. 2024). Many of the large negative biases occur in areas where the SWE exceeds the algorithm detection limit (~150 mm, Sect. 2.1). In Eurasia and central North America, the bias is mainly positive during February 290 and March. Previous work has shown that much of this overestimation is due to the constant snow density exceeding the true snow density in these regions until mid-March (Mortimer et al. 2022, Venäläinen et al. 2023), leading to an overestimation of SWE in these areas. The variable snow density applied in SCv3.1 (Sect. 2.1) reduces much of this positive bias (Venäläinen et al., 2023). In April and May, the bias is primarily negative across the entire Northern Hemisphere.

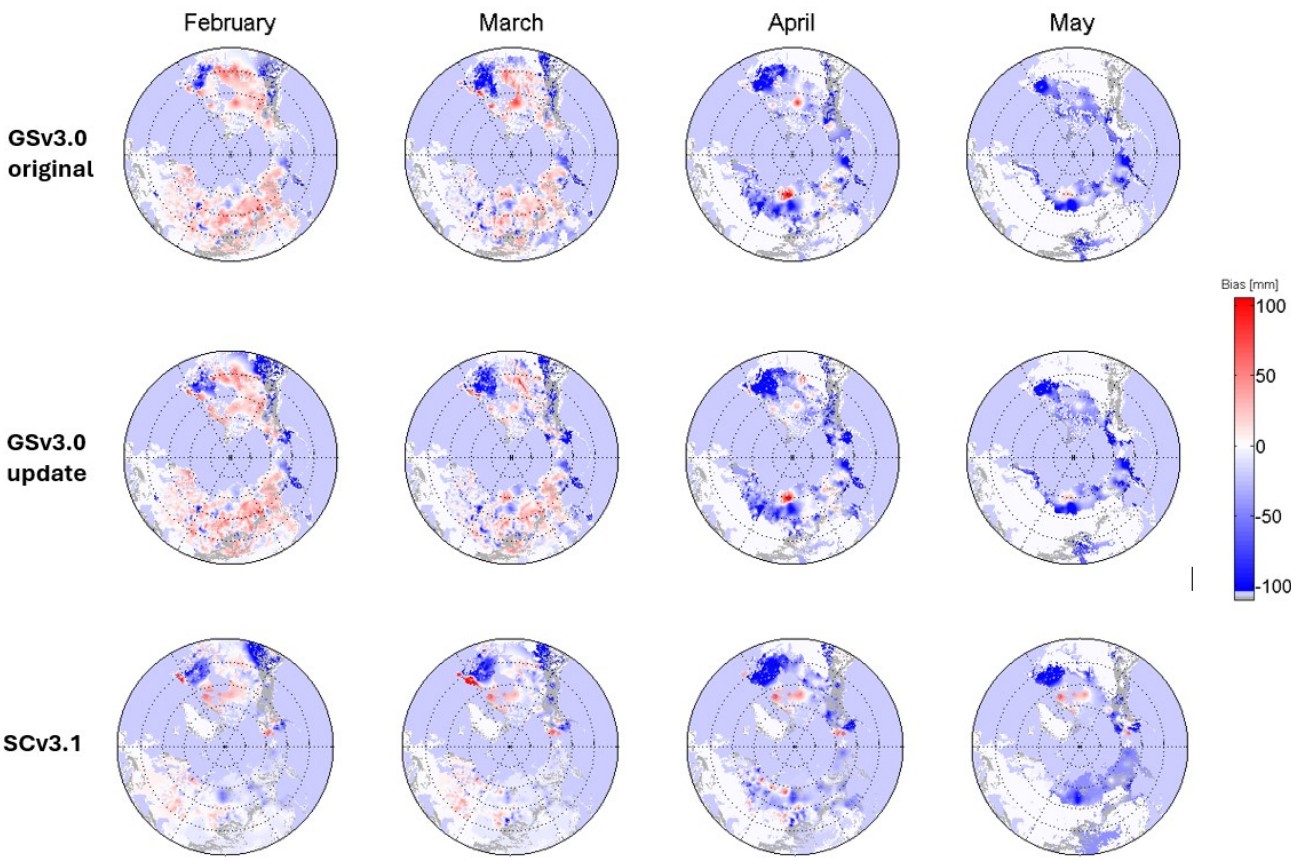

**Figure 4: Monthly bias for February-May calculated for GSv3.0 using the original (top) and updated (middle) reference data, and for SCv3.1 (bottom) with the updated reference data for the mean monthly 1980-2018 snow covered area calculated from the snow extent maps used in SWE retrieval (Sect. 2.1). In practice, the bias correction is applied exclusively to snow-covered areas.**

Although the spatial patterns of bias are similar for both versions of GSv3.0, there are some notable differences in the bias fields (Appendix C). Overall, changes are most pronounced in February and March, and differences are larger in North America. A significant amount of new data was added in Alaska, the western and northeastern US mountains, as well as parts of Quebec and northern Manitoba (Figure 1). In Alaska, these additional data resulted in a larger magnitude and mostly more expansive negative biases for all months except May, when the differences are minimal. Positive biases remain visible in parts of Alaska in February and March, and in parts, original bias correction even results in more snow. In Quebec, the addition of new data reduced the magnitude of positive bias in the northwest (along Hudson Bay) in February. In April and May, the magnitude of the negative bias is larger in the updated fields, whereas in March, it is lower.. In central parts of North America, positive bias observed in the original bias fields is reduced, even becoming negative in some areas during February and March. Finally, despite the addition of new sites in Finland, the bias field remains similar, suggesting that the original snow course data adequately sampled the snow conditions across Finland at the scale of the GlobSnow and Snow CCI products. The high accuracy of SWE retrievals over Finland, due in part to the dense synop SD coverage, may also contribute to the small biases (and hence little change in the bias) in this region.

The impact of the additional reference data is also evident in the regional and hemispheric March snow mass (Figure 5). The Northern Hemisphere March (non-mountainous) snow mass is consistently higher with the updated bias correction (blue line, original bias correction in black). This increased snow mass is attributed mainly to changes in North America (larger negative bias in Alaska and smaller positive biases in central parts of the continent), where the snow mass from the updated bias correction (blue line) is >100 Gt larger than with calculated the original reference data (black line). In

Eurasia, the updated bias correction yields marginally higher snow mass estimates (Figure 5).

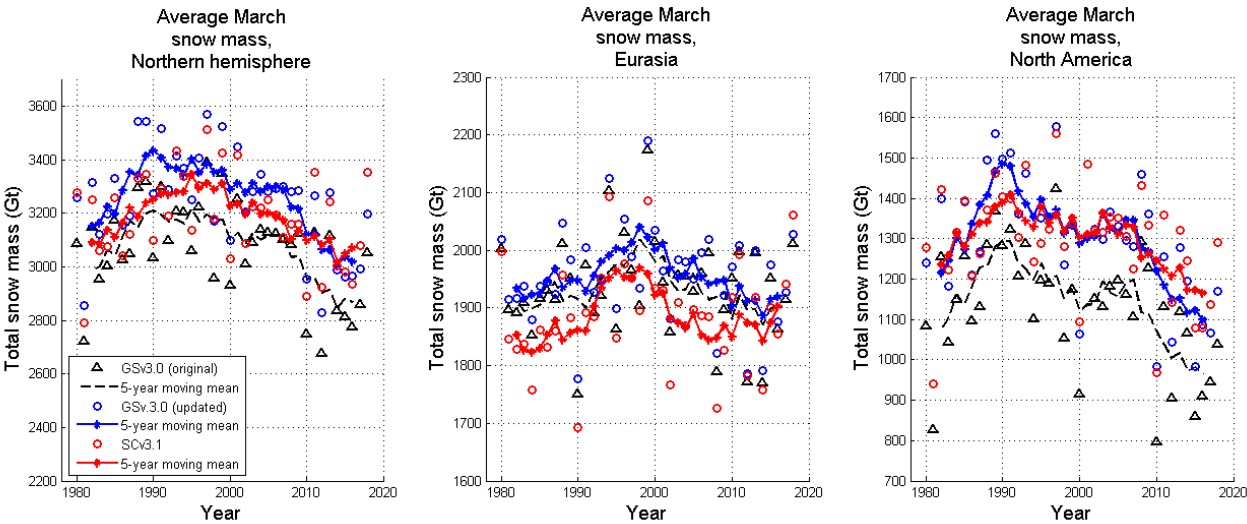

**Figure 5: Mean March snow mass (excluding complex terrain) for the Northern Hemisphere, Eurasia and North America based of the bias-corrected GSv3.0 (black), updated bias-corrected GSv3.0 (blue) and bias-corrected SCv3.1 (red) products with the 5-year running mean (solid lines).**

## 3.2 Impact of changes in SWE retrieval and input data on the bias correction - Snow CCI v3.1 monthly bias correction

Having assessed the impact of altering the snow course dataset on the spatial biases, we apply the updated snow course dataset to an updated version of the GSv3.0 product – SCv3.1. As outlined in Sect. 2.4, observed differences in bias

fields reflect changes made to the retrieval algorithm and input data described in Section 2.3. In general, the magnitude of the bias in SCv3.1 is smaller compared to GSv3.0, particularly across Eurasia and to a lesser extent over central North America (Figure 4), consistent with known improvements to the CCI SWE retrieval (Venäläinen et al., 2023, Mortimer et al., 2022).

In Eurasia, which saw significant changes to the bias field compared to GSv3.0, SCv3.1 has predominantly negative

(positive) biases in Western (Eastern) Eurasia during February and March (Figure 4, bottom row). GSv3.0 has a mostly positive bias in February and a more varied pattern in March. In North America, positive biases in the centre of the continent are reduced during February and March (compared to GSv3.0), even becoming negative in the south-central snow-covered regions during March. In April and May all biases are mostly negative with few local exceptions.

Despite improvements to the SWE retrieval and input data, reflected in the smaller biases compared to GSv3.0, there

are locations and times of the year where the accuracy cannot be improved by tuning parameters because SWE consistently exceeds the retrieval's detection limit (~150-200 mm). In these cases, the bias is consistently negative. This

issue is exemplified by the persistent large negative biases in Quebec and Ontario (Canada), as well as in the western US mountains. Many of these areas also had new snow course sites, which further increased the extent and magnitude of the negative bias (in GSv3.0 compared to that calculated with the original data (Sect 3.1)).

Applying the updated snow course data for both GSv3.0 and SCv.31, we show, in Figure, 6 differences in the 39-year mean monthly SWE for each pixel for the bias corrected monthly GSv3.0 and SCv3.1 products. Red (blue) indicates areas where the bias corrected GSv3.0 product has more (less) SWE compared to the bias corrected SCv3.1 product. Differences are most pronounced, albeit less expansive, in May when the snow extent is smallest. In North America, the updated GSv3.0 tends to have more snow along the coast of Hudson Bay and across much of the prairies during

February and March, as well as along the Alaskan coasts. The SCv3.1 has more snow in eastern North America and across much of the boreal forest, with some exceptions. The differences across Eurasia are more mixed. GSv3.0 has slightly more snow in north-eastern Siberia during February and March, while SCv3.1 has slightly more snow in western Eurasia, with some localised exceptions ( around snow course sites). In Eurasia, there are localised areas with large differences in SWE, notably around the Kara Sea, which has large positive biases in GSv3.0 (both sets of

reference data) but not in SCv3.1 and northeast Siberia and around Ural mountains, which have large negative biases in both GlobSnow products but negligible biases in Snow CCI product (Figure 3).

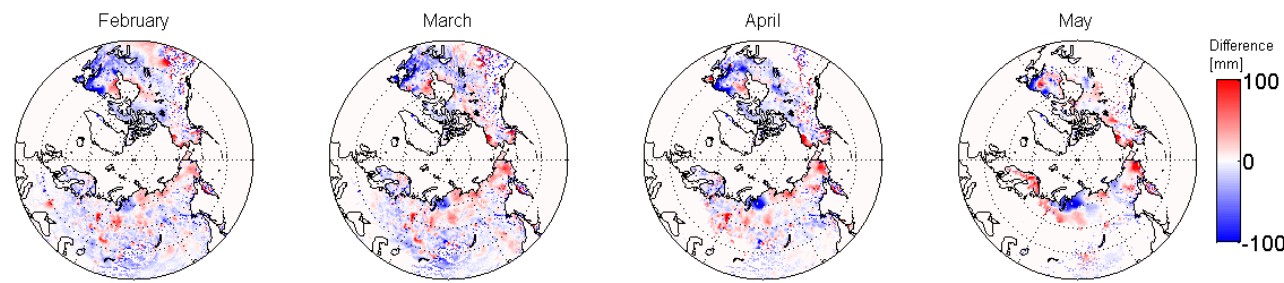

**Figure 6: Difference between 39 year mean monthly bias-corrected GSv3.0 and SCv3.1 products, both using the updated reference data.**

In terms of the time series of March snow mass (Figure 5), when the updated snow course data are used, the SCv3.1 estimates (red line) are consistently lower than those of GSv3.0 (blue line). Most of the hemispheric-scale reduction is attributed to lower Eurasia snow. Changes made to the retrieval (see Sect. 2.2, Mortimer et al., 2022; Venäläinen et al., 2023) reduced the March snow mass in Eurasia by around 100 Gt. Although the bias correction adds snow to Eurasia,

the bias corrected SCv3.1 still has less snow than the bias corrected GSv3.0 product. In North America the spatial differences observed in Figure 6 tend to average out at the continental scale. Except for a few anomalous years (in the lates 1980s), likely tied to changes in PMW Tb data (see Mortimer et al. 2022), the March North American snow mass is similar in GSv3.0 and SCv3.1 (when the same updated reference data are used to calculate the bias).

To place the monthly bias corrected products into a broader context, we compare their respective climatological snow

mass to that of reanalysis products analysed in the SnowPEx project , as described in Sect. 2.6. The updated bias corrected SCv3.1 product shows a clear improvement compared to the original GSv3.0 bias corrected product (Figure 7). The GSv3.0 bias corrected (Figure 7 red crosses) May snow mass is well outside (above) the range estimated by both ensembles and the April snow mass is at the high end of the SnowPEx+ ensemble (blue shading). GSv3.0 February

snow mass is also near the low end of SnowPEx+ spread. The monthly GSv3.0 product was only thoroughly evaluated
for March (Sect. 1; Pulliainen et al. 2020, Luojus et al. 2021), and, as evidenced by Figure 7, the monthly SWE
provided for April and May are clearly too high and for February estimate is quite low. The updated monthly bias
corrected SCv3.1 product is a clear improvement with its monthly climatological SWE (Figure 7, grey square) falling in
the middle of the range estimated by the SnowPEx+ product suite.

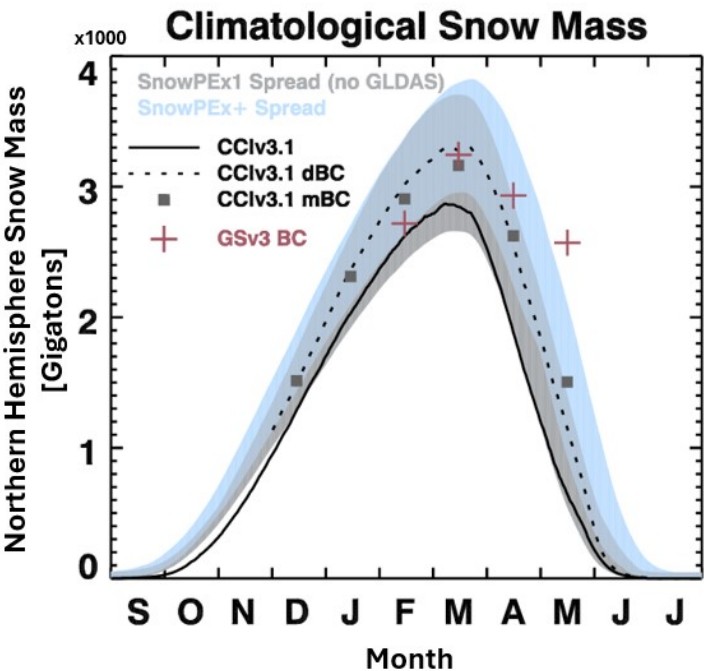

**Figure 7: Northern hemisphere climatological snow mass 1980 – 2018, excluding complex terrain. Shading shows the range of products included in the SnowPEx (grey) and SnowPEx+ (blue) Intercomparison projects. Crosses indicate values from the GSv3.0 bias corrected monthly product and squares show SCv3.1 bias corrected monthly product. Dashed line shows the daily bias corrected SCv3.1 and solid line is 'non-bias-corrected' SCv3.1 product.**

### 3.3 Daily bias correction

We computed daily bias maps for each day from December through May using the monthly SCv3.1 bias correction
maps (Sect. 2.2.2). These daily bias maps were then used to bias correct SCv3.1 product between 1980-2022. To
understand the impact of the bias correction on the accuracy of the daily product, we compare the daily SCv3.1
uncorrected and bias-corrected products to airborne gamma SWE observations (Table 2) and a suite of reanalysis
products (SnowPEx products, Sect. 2.5). Supplemental evaluation with the same snow course data used to calculate the
bias maps is provided in Table D1.

**Table 2: Validation parameter for SWE < 500 mm/SWE < 200 mm for North America for 1980-2022 calculated using independent airborne gamma SWE measurements.**

|  | RMSE [mm] | MAE [mm] | Bias [mm] | Correlation Coefficient |
|---|---|---|---|---|
| GlobSnow v3.0 | 51.9/42.3 | 37.0/31.9 | -20.7/-14.7 | 0.56/0.52 |
| SnowCCI v3.1 | 48.7/41.8 | 35.4/31.7 | -18.8/-14.3 | 0.65/0.60 |
| SnowCCI v3.1, bias corrected | 45.9/43.1 | 32.0/30.2 | -3.3/-0.4 | 0.68/0.60 |

Based on comparisons with airborne gamma SWE validation (Table 2), daily bias correction results in a large improvement in the bias and marginal improvement in the MAE for both upper SWE limits (< 200 mm and < 500 mm). For SWE < 500 mm, the RMSE and correlation also improved slightly. For the lower SWE limit (< 200 mm), RMSE degraded slightly for the bias-corrected product, and there is no change in the correlation. Importantly, however, the airborne gamma SWE data are restricted to the US and parts of southern Canada (Figure 3), so the corresponding validation may not be indicative of the product's hemispheric performance. Notably, it excludes much of the high SWE areas in the northern boreal forest, which tends to have high SWE and large biases in the uncorrected product (Figure 4). Most tundra regions and all of Eurasia are also excluded from this validation dataset. Therefore, we also calculated validation statistics with the snow course data (Table D1), despite the aforementioned caveats (Sect. 2.5). Since these data are used to perform the bias correction and to inform the snow density used within the retrieval, we expect strong agreement between the bias-corrected CCI data and the snow course observations. As expected from Figure 4 and Sect. 2.3-2.5, the impact of the bias correction is greater for North America compared to Eurasia. Further, despite applying a bias correction, the RMSE and MAE are still considerably larger in North America.

To extend our evaluation across the full Northern Hemisphere snow-covered area, we compare the daily SWE fields to those of a suite of reanalysis products (Sect. 2.5), as shown in Figure 8. The comparison includes data from all months between December and May. Figure 8 also includes a comparison of the bias-corrected and original SCv3.1 products. It is important to note that while the ensemble of reanalysis products provides reasonable SWE estimates, the ensemble does not represent ground truth values. Some differences observed between the reanalysis products and SCv3.1 may reflect limitations in the reanalysis datasets.

Consistent with Figures 3 and 5, the bias correction increases the Northern Hemisphere snow mass compared to the original SCv3.1 product, with the largest changes occurring in April and May. Regionally, significant increases are seen in eastern Canada and in areas bordering the complex topography mask across all months (Figure 8).

Overall, compared to the reanalysis mean, the bias-corrected product has more snow in North America, arctic regions excepted, and less snow in Eurasia, mountainous regions excepted. In detail, bias-corrected SCv3.1 has less snow mass across western and northern Eurasia, with some exceptions, and more snow mass in Finland (May excepted), southern Eurasia, and in mountainous areas or those bordering the complex topography mask. In North America, there are notable areas with considerably higher SWE (~40-60 mm higher) than the reanalysis mean in south-west Quebec, Canada and in areas bordering the complex topography mask in the west. There is generally less snow mass in the eastern Arctic [North America] and areas bordering Hudson Bay. Elsewhere, the bias-corrected SCv3.1 has higher SWE than the reanalysis mean.

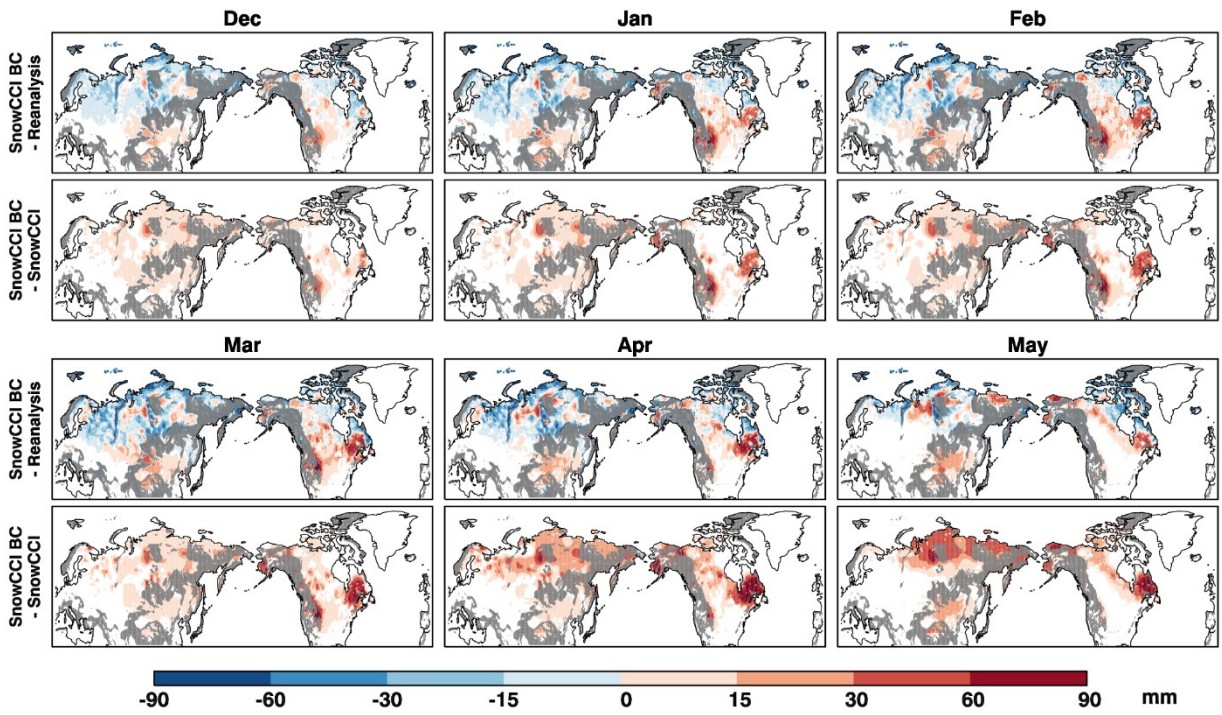

**Figure 8. Pixel-wise comparison of the monthly average of the daily bias corrected SCv3.1 and a suite of reanalysis and comparison of SCv3.1 and bias corrected SCv3.1 for December to May. Masked complex terrain is shown with grey.**

Finally, although the improvement in product accuracy captured by the comparisons with in situ data is small, there is a large improvement in the Northern Hemisphere climatological snow mass estimation. The uncorrected SCv3.1 product is at the bottom of the SnowPex+ suite and at the low end of the SnowPex1 suite. The bias correction adds (~500 Gt) snow mass such that its climatological SWE is in the middle of the reanalysis product spread.

For the daily bias corrected product (Figure 7, dashed lines), the peak amount of snow is about 500 Gt larger than for the non-corrected products. This increase in snow mass brings the peak snow mass closer to the snow mass estimates of reanalysis products (Mortimer et al. 2022), and as seen in Figure 7, the bias corrected peak snow mass is close to the middle of the spread instead of near the lower end.

## 4 Discussion

A key limitation of passive microwave SWE retrievals is their systematic underestimation of large SWE values. These retrievals rely on differences in measured Tb between frequencies sensitive to snow grain volume scattering and those insensitive to snow (Chang et al., 1987; Kelly, 2009; Tedesco et al., 2010). When snow depth is substantial (SWE ~ > 150 mm), the snowpack transitions from a scattering medium to a source of emission, leading to the underestimation of large SWE values. Assimilating in situ snow depth data, as implemented in the GlobSnow SWE retrieval, partially mitigates this issue and enhances estimates of moderate snowpacks (SWE $\sim<$ 200 mm) (Mortimer et al., 2020). However, as illustrated in Figure 6, the underestimation of large SWE values still persists in both GlobSnow SWE and updated Snow CCI SWE retrievals. Based on the findings of this study, daily bias correction presents a promising approach to address this underestimation problem.

Daily bias correction adds a notable amount of snow (~500 Gt) to the Northern Hemisphere climatological snow mass, bringing the bias-corrected values consistent with those of reanalysis and model-based products (Figure 8). This improvement is important for analysing long-term and large-scale trends in snow mass. Based on validation with airborne gamma SWE (Table 1), the estimation of large SWE values is also improved with daily bias correction. This is

expected because the physics of the retrieval method limits the uncorrected values to shallow and moderate snowpacks. In areas of high SWE, in most cases, the applied bias correction adds SWE to bring the estimate closer to the true value. Although the Hemispheric SWE is clearly improved, there are notable regional differences, as illustrated through our comparison with reanalysis data. In the future, comparison with region specific reanalysis data might help to identify region where SWE retrieval is most inaccurate.

As discussed, global validation of bias corrected products is challenging due to a lack of independent reference data, but validation was performed using airborne gamma data available over the US and southern Canada (Table 1). As presented in Luojus et al. 2021, when assessed with the same snow courses used to produce the monthly bias correction, the bias of the uncorrected data is roughly equivalent to that of the corrected product, less the value of the bias correction field. Given the dependence of the bias correction on the snow course data, it is not surprising that validation

statistics obtained using those data outperform those based on the airborne gamma data. For example, central North America is well covered by airborne gamma but not by snow courses which are used to develop the bias correction. Consequently, the larger errors obtained when assessed with airborne gamma partly reflect the inability of the bias correction to correct biases in areas with limited in situ information. This highlights the limitations of the bias correction in regions with sparse or no in situ data. Unfortunately, since the airborne gamma data do not cover all snow classes or

the full winter season, we are unable to discern whether the magnitude of the errors obtained with airborne gamma apply to other regions

Additionally, the difference in the timing and SWE distribution of the two validation datasets may also contribute to the differing accuracies when calculated using snow course and airborne gamma. Previous work (Mortimer et al. 2022 Figure 6) has shown that errors in the SCv3.1 product increase over the course of the snow season in concert with SWE

magnitude. Airborne gamma surveys are usually conducted once or twice per year and are concentrated in February and March near peak SWE (Sect. 2.6). Snow courses are conducted more frequently (~ 14 to 30 days in North America (see Mortimer et al. 2024)) and tend to cover the full snow season. This means that snow courses are more likely to capture lower the SWE values of the accumulation season which are often associated with smaller absolute errors (see for example, Mortimer et al. 2020). In contrast, the airborne gamma observations are biased towards higher values during

the middle and end of the snow season which are both associated with larger absolute errors. To demonstrate, Figure E1 shows that the SWE distribution of the airborne gamma is shifted higher compared to snow courses, although the snow course data cover a much larger range.

The addition of new reference data has a greater effect on bias in North America than in Eurasia. This is expected as more data were added to North America and the Eurasia dataset remained similar, Finland excepted. Additional data

made the bias more negative in Quebec and Alaska and reduced the extent and magnitude of the positive bias in central North America. Together, these changes added ~ 100 Gt to the snow March snow mass (Figure 5). It is notable that in Finland, where the GSv3.0 bias is small and a substantial amount of new data were added, the bias remained similar. This indicates that the original data adequately sampled the snow conditions in this region, which already has very accurate SWE retrievals.

Although the updated reference data had little impact on the Eurasia SWE, there were large changes between the bias-corrected GSv3.0 and SCv3.1 products (both using the updated data). These differences may be attributed to one or more factors, outlined below. Changes to the retrieval algorithm, namely the move from a static snow density (240 kg m$^{-3}$) to spatially and temporally variable values, decreased the snow mass over much of Eurasia, where the true snow density is lower than the static parameter up until March (Venäläinen et al., 2023). This change, which was seen as an

improvement because products using the static density tended to overestimate SWE in Eurasia, is visible in Figure 4. The GSv3.0 bias fields are predominantly positive in Eurasia in February and March, while those of SCv3.1 are slightly negative, and the uncorrected March SWE is lower for SCv3.1 compared to GSv3.0. In April and May static snow density tends to be smaller than the actual snow density and change to dynamic snow density consideration has reduced negative bias in these months.

As detailed in Sect. 2.1, GSv3.0 and SCv3.1 product and bias fields are produced in different resolutions, which can influence bias fields. On hemispheric scale, change to finer grid spacing can provides some improvements to correlation, RMSE and bias for the SWE retrieval but has minimal effect on hemispheric snow mass (Mortimer et al., 2022). However, on local scale different resolutions can produce clear differences. For example, both GSv3.0 and SCv3.1 overestimate SWE around the Kara Sea area in Siberia in March and April, but positive bias is much more

significant for GSv3.0. There are a handful of snow courses where radiometer-based SWE is systematically overestimated, while underestimation is a bigger problem in surrounding locations. The effect of these few locations is bigger for the coarser 25 km grid (GSv3.0) than for the finer 12.5 km grid (SCv3.1), though changes in snow density parameterisation also affect bias in the area. Effects of resolution can also be seen around mountainous areas where GSv3.0 SWE tends to be larger (for example near Ural Mountains and mountains in eastern Siberia and western

Alaska). Complex terrains are masked out from the SWE products, but masks are not identical for the two products.

    Finally, the way the monthly SWE was calculated may also have an impact on the differences in SWE estimates between GSv3.0 and SCv3.1, especially during the ablation season in May. GSv3.0 monthly values were calculated as the arithmetic mean of days with valid SWE observations. This means that days without SWE are not accounted for average SWE calculation. We adjusted the method for SCv3.1 to try to better account for missing SWE retrievals as

outlined in Sect 2.2.1, In May most missing days are towards the end of the season and GSv3.0 monthly estimate is considerably larger than modelled estimates. SCv3.1 monthly estimate is much closer to expected value, partially due to filling missing days and thus reducing the monthly average value. .

    As the bias correction is based on 40 years of data, it may compromise the local accuracy of SWE estimates. If estimates are accurate in some years but inaccurate in others at the same location, bias correction might overcorrect

estimates in years with initially good estimates. When looking at airborne gamma data, the bias corrected SCv3.1 has a higher RMSE value for SWE < 200 mm than the original (uncorrected) product. This is due to bias correction creating outlier values by overcorrecting SWE estimates. Gamma validation shows that bias correction may lead to overestimation of small (< 50 mm) SWE values. Future work could explore temporal or spatial constraints to refine when and where the bias correction is applied. Specifically, the SCv3.1 SWE product includes pixel-wise uncertainty

information for SWE estimates, offering a possible way to target the bias correction to specific locations. Excluding small SWE values from the bias correction could also be considered.

    The bias-correction method studied in the paper assumes that snow course data are uniformly distributed throughout each month. This assumption generally holds during mid-winter. However, in December and January, observations are

weighted toward the end of the month, and in May, toward the beginning of the month. This uneven distribution may influence the monthly bias correction fields. For instance, if the lack of observations later in May reflects an absence of snow, the calculated biases may be overestimated. We have tested how sensitive the bias correction method is to the distribution of reference data by adjusting the temporal centering of the monthly bins. We tested re-centering the bins around the 1st of each month (i.e., using data from the 16th of one month to the 15th of the next). This had little effect in mid-winter months like December and February, but a more noticeable impact in May, where the re-centered bias correction resulted in increased snow mass, which could be interpreted as a degradation in performance. Additionally, we have also tested extrapolating bias data into periods with limited observations at the start of December and end of May. In December, extrapolation slightly increased snow mass estimates, while in May it led to a decrease. These results suggest that while mid-winter bias estimates are robust, bias corrections during the shoulder seasons are affected by the uneven distribution and limited availability of reference data.In this paper simple mean bias subtraction was used as the focus was on updating an existing and proven method and expanding it to new months and to a higher temporal frequency (daily instead of the original monthly). Our results provide a baseline against which bias correction methods could be studied in the future. For example, King et al., 2020 studied different SWE bias correction methods at local scales, finding that random forest techniques provided accurate results, indicating the potential of machine learning-based approaches for bias correction

## 5 Conclusion

In this study, we updated monthly bias correction fields used to improve monthly passive microwave assimilation-based SWE retrievals and snow mass estimates. We updated the fields using snow reference data from new sources and calculated them for the newest assimilation-based SWE retrieval, SCv3.1. Bias correction was also extended to December and January and to a daily time scale.

Updated reference data had a larger effect on bias in North America than changes in the algorithm did. On the other hand, in Eurasia, the addition of updated reference data did not change bias significantly, but changes in the algorithm had a clear effect on bias.,

Daily bias correction added a significant amount of snow to the Northern Hemisphere snow mass estimation, bringing it closer to reanalysis products. Daily bias correction can also provide moderate improvements to SWE retrieval but compromise accuracy on a local scale.

The continued development of the SWE retrieval algorithm remains important. Improvements in uncorrected SWE products are also seen in bias corrected products. For example, improved snow mass peak timing of the SCv3.1 product is also visible on the bias corrected product. Snow mass estimations based on monthly bias corrected SCv3.1 products have improved significantly for April and May in comparison to GSv3.0.

**Appendix A: Evolution of annual bias in SCv3.0 SWE estimates for March.**

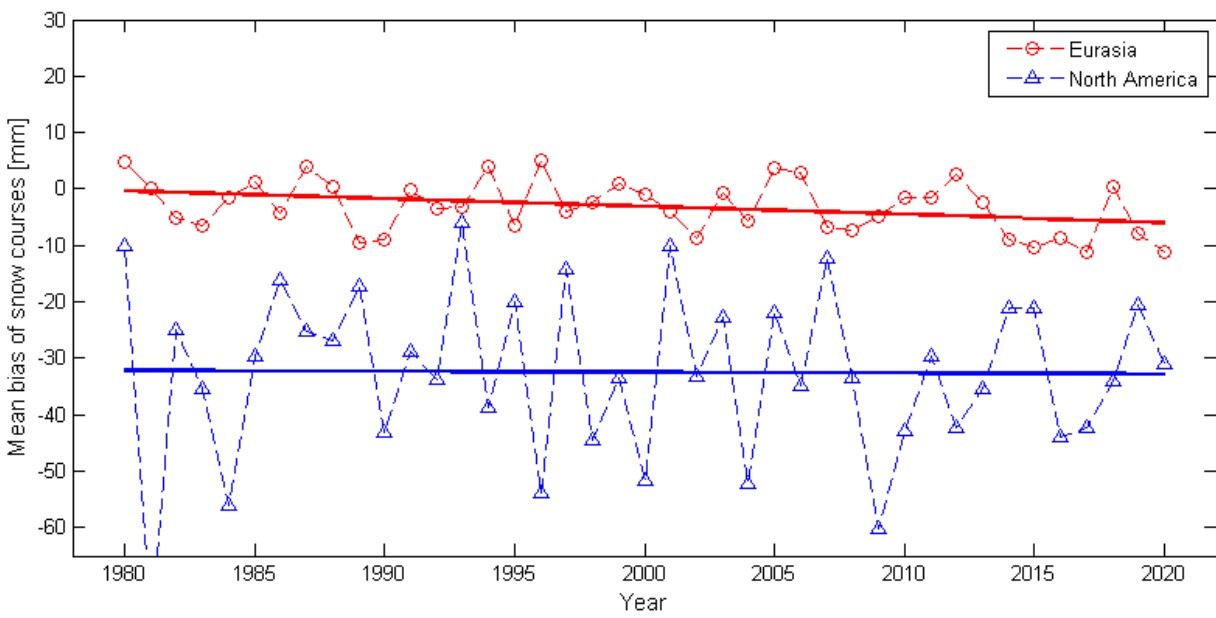

**Figure A1. Mean annual bias of snow courses in Eurasia (red) and North America (blue). A very slight negative trend is visible for Eurasia (p-value 0.055). For North America trend is more visible but still negligible (p-value 0.94).**

**Appendix B: December and January bias maps for SCv3.1**

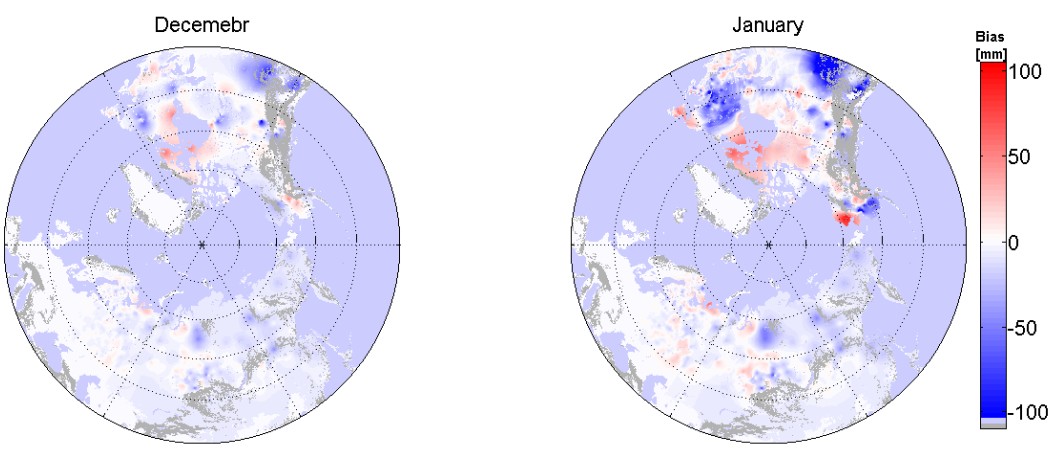

**Figure B1. December and January bias maps for SCv3.1. Bias is small in December for the whole Northern Hemisphere. In January, both positive and negative biases are visible in North America**.

## Appendix C: Comparison of GSv3.0 bias corrected products

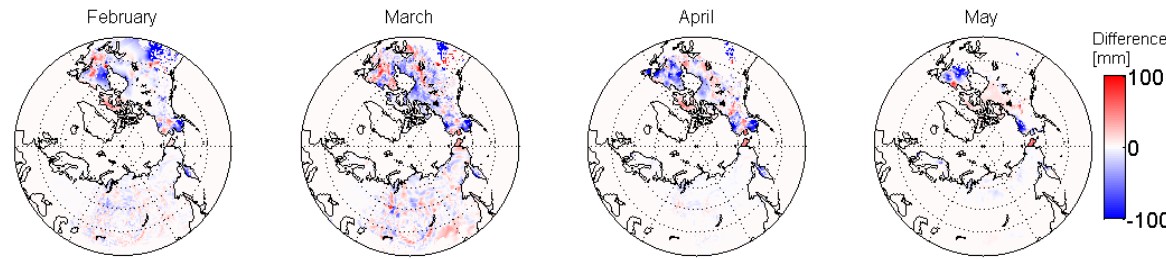

**Figure C1. The difference in monthly mean SWE values between the two bias corrected GSv3.0 products (origanal – updated).**

## Appendix D: Snow course validation

**Table D1. Validation parameter for SWE < 500 mm/SWE < 200 mm for Northern Hemisphere, Eurasia, and North America for 1980-2022**.

| | RMSE [mm] | MAE [mm] | Bias [mm] | Correlation coefficient |
|---|---|---|---|---|
| GlobSnow v3.0 | 50.3/37.3 | 33.4/27.7 | -6.8/0.74 | 0.64/0.67 |
| SnowCCI v3.1 | 46.4/36.2 | 29.3/24.8 | -11.6/-6.4 | 0.73/0.74 |
| SnowCCI v3.1, bias corrected | 37.7/32.9 | 25.1/22.9 | 3.7/6.4 | 0.83/0.80 |
| **Eurasia** | | | | |
| GlobSnow v3.0 | 39.6/33.0 | 27.7/25.1 | 1.0/4.8 | 0.73/0.74 |
| SnowCCI v3.1 | 36.8/31.8 | 23.8/21.7 | -6.2/-3.8 | 0.79/0.79 |
| SnowCCI v3.1, bias corrected | 33.5/29.9 | 21.9/20.5 | 3.7/5.3 | 0.83/0.82 |
| **North America** | | | | |
| GlobSnow v3.0 | 77.2/51.1 | 53.8/38.5 | -34.5/-15.7 | 0.53/0.52 |
| SnowCCI v3.1 | 67.0/47.8 | 45.6/34.9 | -27.6/-14.9 | 0.65/0.60 |
| SnowCCI v3.1, bias corrected | 47.3/40.6 | 33.7/29.7 | 3.8/9.2 | 0.82/0.75 |

## Appendix E: Distribution of reference SWE measurements

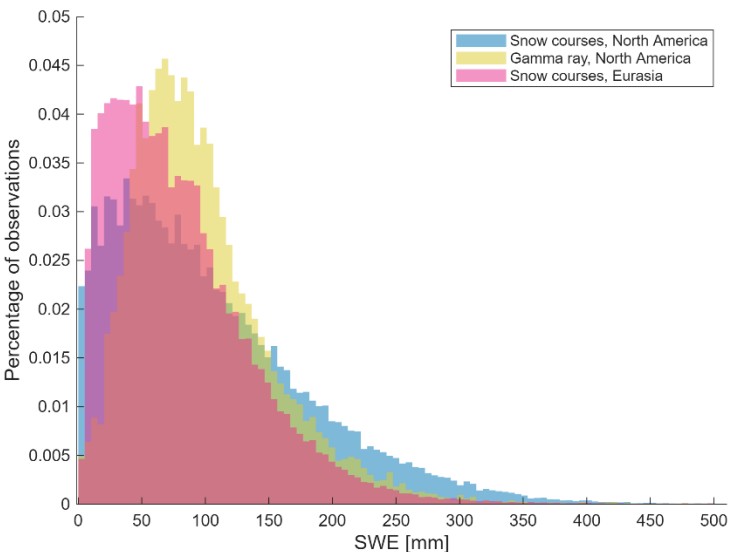

**Figure E1: Normalized distribution of SWE measurements for snow course data from North America (red) and Eurasia (green) and from gamma ray data from North America (blue). Observations from areas of complex terrain are removed. Percentages are calculated for each set individually. Gamma ray dataset has less small SWE values than snow course sets.**

**Code availability.** The GlobSnow code is available at: http://www.globsnow.info/swe/archive_v3.0/source_codes/

**Data availability.** GlobSnow v3.0 data is available at: https://www.globsnow.info/swe/archive_v3.0/L3A_daily_SWE/. Snow CCI v3.1 is available at: https://catalogue.ceda.ac.uk/uuid/9d9bfc488ec54b1297eca2c9662f9c81/

**Author contribution**. P.V., K.L., and J.P. conceived the concept of the study; P.V. performed the data processing and produced the first draft of the manuscript, which was subsequently edited by C.M. and K.L.; P.V., C.M. and L.M. performed data analysis; M.T. contributed to the data collection, analytical tools, and methods.

**Competing interests**. The authors declare that they have no conflict of interest.

**Acknowledgement.** This work is supported by the ESA CCI+ Snow project (4000124098/18/I-NB).

**Sources**

Armstrong, R., Knowles, K., Brodzik, M., and Hardman, M. A.: DMSP SSM/I-SSMIS Pathfinder Daily EASE-Grid Brightness Temperatures, Version 2, NASA DAAC at the National Snow and Ice Data Center, Boulder, Colorado, USA, available at: https://nsidc.org/data/NSIDC-0032/versions/2 (last access: 26 May 2021), 1994.

Brodzik, M. J., Long, D. G., Hardman, M. A., Paget, A., and Armstrong, R. L.: MEaSUREs Calibrated EnhancedResolution Passive Microwave Daily EASE-Grid 2.0 Brightness Temperature ESDR, Version 1, National Snow and Ice Data Center, Boulder, Colorado, USA [data set], https://doi.org/10.5067/MEASURES/CRYOSPHERE/NSIDC0630.001, 2016.

Brown, R., Tapsoba, D., and Derksen, C.: Evaluation of snow water equivalent dataset over the Saint-Maurice river basin region of southern Quebec, Hydrol. Process., 32: 2748–2764, https://doi.org/10.1002/hyp.13221, 2018.

Bulygina, O.N., Groisman, P.Y., Razuvaev, V.N., and Korshunova, N.N.: Changes in snow cover characteristics over Northern Eurasia since 1966, Environmental Research Letters 6, 045204, https://doi.org/10.1088/1748-9326/6/4/045204, 2011.

Carroll, T. R.: Airborne Gamma Radiation Snow Survey Program: A user's guide, Version 5.0, National Operational Hydrologic Remote Sensing Center (NOHRSC), Chanhassen, 14, 2001.

Chang, A.T.C., Foster, J.L., and Hall, D.K.: Nimbus-7 SMMR Derived Global Snow Cover Parameters, Ann Glaciol 9, 39–44, https://doi.org/10.3189/S0260305500200736, 1987.

Cho, E., Jacobs, J. M., and Vuyovich, C. M.: The Value of Long-Term (40 years) Airborne Gamma Radiation SWE Record for Evaluating Three Observation-Based Gridded SWE Data Sets by Seasonal Snow and Land Cover Classifications, Water Resources Research, 56, e2019WR025813, https://doi.org/10.1029/2019WR025813, 2020.

Derksen, C., and Brown, R.: Spring snow cover extent reductions in the 2008-2012 period exceeding climate model projections, Geophys Res Lett, 39, https://doi.org/10.1029/2012GL053387, 2012.

Derksen, C., Nagler, T., and Schwaizer, G.: ESA CCI+ Snow ECV: User Requirements Document, version 4.0, May 2022.

Derksen, C., Walker, A., and Goodison, B.: Evaluation of passive microwave snow water equivalent retrievals across the boreal forest/tundra transition of western Canada, Remote Sens Environ, 96, 315–327, https://doi.org/10.1016/j.rse.2005.02.014, 2005.

GCOS: The 2022 GCOS ECV requirements. GCOS-245, World Meteorological Organization, 244 pp., https://library.wmo.int/records/item/58111-the-2022-gcos-ecvs-requirements-gcos-245, 2022.

Haberkorn, A.: European Snow Booklet – an Inventory of Snow Measurements in Europe, EnviDat, https://doi:10.16904/envidat.59, 2019.

Hall, N. D., Stuntz, B. B., and Abrams, R. H. Climate Change and Freshwater Resources, Nat. Resour. Environ., 22, 30–35, 2008.

Hall, D. K., Kelly, R. E. J., Riggs, G. A., Chang, A. T. C., and Foster, J. L.: Assessment of the relative accuracy of hemispheric-scale snow-cover maps, *Ann. Glaciol.*, 34, 24–30, https://doi.org/10.3189/172756402781817770, 2002.

Hori, M., Sugiura, K., Kobayashi, K., Aoki, T., Tanikawa, T., Kuchiki, K., and Enomoto, H.: A 38-year (1978–2015) Northern Hemisphere daily snow cover extent product derived using consistent objective criteria from satellite-borne optical sensors, *Remote Sens. Environ.*, 191, 402–418, https://doi.org/10.1016/j.rse.2017.01.023, 2017.

Jones, H. G., Pomeroy, J. W., Walker, D. A., and Hoham, R. W.: Snow Ecology: An Interdisciplinary Examination of SnowCovered Ecosystems, Cambridge University Press, 2011.

Kelly, R.: The AMSR-E Snow Depth Algorithm: Description and Initial Results, J. Remote Sens. Soc. JPN, 29, 307–317, https://doi.org/10.11440/rssj.29.307, 2009.

Kelly, R., Chang, A., Tsang, L., Foster, J.: A prototype AMSR-E global snow area and snow depth algorithm, IEEE Transactions on Geoscience and Remote Sensing 41, 230–242, https://doi.org/10.1109/TGRS.2003.809118, 2003.

King, F., Erler, A. R., Frey, S. K., and Fletcher, C. G.: Application of machine learning techniques for regional bias correction of snow water equivalent estimates in Ontario, Canada, Hydrology and Earth System Sciences, 24(10), 4887–4902, 2020.

Kitaev, L., Kislov, A., Krenke, A., Razuvaev, V., Martuganov, R., & Konstantinov, I:. The snow cover characteristics of northern Eurasia and their relationship to climatic parameters. Boreal Environment Research, 7, 437-445, ISSN:1797-2469, 2002.

Knowles, K., Njoku, G., Armstrong, R., and Brodzik, M.: Nimbus7 SMMR Pathfinder Daily EASE-Grid Brightness Temperatures, version 1, NASA National Snow Ice Data Center Distributed Active Archive Center, Boulder, Colorado, USA, https://doi.org/10.5067/36SLCSCZU7N6, 2000

Kuusisto, E.: Snow accumulation and snowmelt in Finland, vol. 55, Water Research Institute, Helsinki, 149 pp., ISBN 951-46-7494-4, 1984.

Larue, F., Royer, A., De Sève, D., Langlois, A., Roy, A., and Brucker, L.: Validation of GlobSnow-2 snow water equivalent over Eastern Canada, Remote Sens. Environ., 194, 264–277, https://doi.org/10.1016/j.rse.2017.03.027, 2017

Luojus, K., Pulliainen, J., Takala, M., Lemmetyinen, J., Mortimer, C., Derksen, C., Mudryk, L., Moisander, M., Hiltunen, M., Smolander, T., Ikonen, J., Cohen, J., Salminen, M., Norberg, J., Veijola, K., Venäläinen, P.: GlobSnow v3.0 Northern Hemisphere snow water equivalent dataset, Sci Data, 8, https://doi.org/10.1038/s41597-021-00939-2, 2021.

Magnusson, J., Nævdal, G., Matt, F., Burkhart, J. F., and Winstral, A.: Improving hydropower inflow forecasts by assimilating snow data, Hydrol. Res., 51, 226–237, https://doi.org/10.2166/nh.2020.025, 2020.

Mortimer, C., Mudryk, L., Derksen, C., Brady, M., Luojus, K., Venäläinen, P., Moisander, M., Lemmetyinen, J., Takala, M., Tanis, C., Pulliainen, J.: Benchmarking algorithm changes to the Snow CCI+ snow water equivalent product, Remote Sens Environ, 274, 112988, https://doi.org/10.1016/j.rse.2022.112988, 2022.

Mortimer, C., Mudryk, L., Derksen, C., Luojus, K., Brown, R., Kelly, R., Tedesco, M.: Evaluation of long-term Northern Hemisphere snow water equivalent products, Cryosphere 14, 1579–1594, https://doi.org/10.5194/tc-14-1579-2020, 2020.

Mortimer, C., and Vionnet, V.: Northern Hemisphere Historical In-situ Snow Water Equivalent Dataset (1979-2021), 1, Zenodo, doi:10.5281/zenodo.10287093, 2024.

Mortimer, C., Mudryk, L., Cho, E., Derksen, C., Brady, M., and Vuyovich, C.: Use of multiple reference data sources to cross-validate gridded snow water equivalent products over North America, The Cryosphere, 18, 5619–5639, https://doi.org/10.5194/tc-18-5619-2024, 2024.

.

Mudryk, L.R., Derksen, C., Kushner, P.J., Brown, R.: Characterization of Northern Hemisphere Snow Water Equivalent Datasets, 1981–2010, J Clim 28, 8037–8051, https://doi.org/10.1175/JCLI-D-15-0229.1, 2015.

Mudryk, L., Mortimer, C., Derksen, C., Elias Chereque, A., Kushner, P.: Benchmarking of SWE products based on outcomes of the SnowPEx+ Intercomparison Project, EGUsphere [preprint], https://doi.org/10.5194/egusphere-2023-3014, 2024.

Mätzler, C.: Passive microwave signatures of landscapes in winter, Meteorology and Atmospheric Physics, 54, 241–260, https://doi.org/10.1007/BF01030063, 1994.

Pulliainen, J.: Mapping of snow water equivalent and snow depth in boreal and sub-arctic zones by assimilating space-borne microwave radiometer data and ground-based observations, Remote Sens Environ, 101, 257–269, https://doi.org/10.1016/j.rse.2006.01.002, 2006.

Pulliainen, J., Grandell, J., Hallikainen, M.: HUT snow emission model and its applicability to snow water equivalent retrieval, IEEE Transactions on Geoscience and Remote Sensing, 37, 1378–1390, https://doi.org/10.1109/36.763302, 1999.

Pulliainen, J., Luojus, K., Derksen, C., Mudryk, L., Lemmetyinen, J., Salminen, M., Ikonen, J., Takala, M., Cohen, J., Smolander, T., Norberg, J.: Patterns and trends of Northern Hemisphere snow mass from 1980 to 2018, Nature 581, 294–298, https://doi.org/10.1038/s41586-020-2258-0, 2020.

Serreze, M.C., Clark, M.P., Armstrong, R.L., McGinnis, D.A., Pulwarty, R.S.: Characteristics of the western United States snowpack from snowpack telemetry (SNOTEL) data, Water Resour Res 35, 2145–2160, https://doi.org/10.1029/1999WR900090, 1999.

Solberg, R., Killie, M.A., Andreassen, L.M., and König, M.: CryoClim: A new system and service for climate monitoring of the cryosphere. IOP Conference Series: Earth and Environmental Science, vol. 17, no. 1, p. 012008. IOP Publishing, 2014.

Takala, M., Luojus, K., Pulliainen, J., Derksen, C., Lemmetyinen, J., Kärnä, J.P., Koskinen, J., Bojkov, B.: Estimating northern hemisphere snow water equivalent for climate research through assimilation of space-borne radiometer data and ground-based measurements, Remote Sens Environ 115, 3517–3529, https://doi.org/10.1016/j.rse.2011.08.014, 2011.

Takala, M., Pulliainen, J., Metsamaki, S. J., and Koskinen, J. T.: Detection of snowmelt using spaceborne microwave radiometer data in Eurasia from 1979 to 2007, *IEEE Trans. Geosci. Remote Sens.*, 47, 2996–3007, https://doi.org/10.1109/TGRS.2009.2018442, 2009

Tedesco, M. and Narvekar, P. S.: Assessment of the NASA AMSR-E SWE Product, IEEE J. Sel. Top. Appl. Earth Obs. Remote Sens., 3(1), 141-159, https://doi.org/10.1109/JSTARS.2010.2040462, 2010.

Venäläinen P, Luojus K, Mortimer C, Lemmetyinen J, Pulliainen J, Takala M, Moisander M, Zschenderlein L.: Implementing spatially and temporally varying snow densities into the GlobSnow snow water equivalent retrieval. The Cryosphere, 17(2), https://doi.org/10.5194/tc-15-2969-202,2023.

Vionnet, V., Mortimer, C., Brady, M., Arnal., L., Brown, R.: Canadian historical Snow Water Equivalent dataset (CanSWE, 1928–2020), Earth Syst. Sci. Data, 13, 4603–4619, https://doi.org/10.5194/essd-13-4603-2021, 2021.

Zschenderlein L, Luojus K, Takala M, Venäläinen P, Pulliainen J.: Evaluation of passive microwave dry snow detection algorithms and application to SWE retrieval during seasonal snow accumulation. Remote Sensing of Environment, 288, https://doi.org/10.1016/j.rse.2023.113476,2023.