# Peer review of "Updated monthly and new daily bias correction for assimilation-based passive microwave SWE retrieval"

_EGUsphere, 2024_

## Author Comment (AC1)

Reply to reviewer #2

*We thank the reviewer for their time and constructive comments on the manuscript. We take all the comments and suggestions into account. Our replies are written in italic font.*

General comment:

The Results section is somewhat confusing and its structure should be revised. I suggest first discussing the temporal evolution of the bias and then its spatial heterogeneity. Then I suggest comparing the original and updated versions of the GlobSnow V3.0 and finally compare with the SCv3.1 bias field. Perhaps, it would also be helpful to present a table in the Methods section with the different products you are comparing.

*We agree that the Results section was a bit cumbersome. In our current structure, we first analyze the effects of updating the snow course data applied to GSv3.0, then the effects of changes in the retrieval algorithm (GSv3.0 vs SCv3.1), and finally we validate and evaluate the new daily bias-corrected SCv3.1 product. To make it easier for readers to follow this structure, we have added an explanation in Methods Section 2.6 and renamed one subsection in the Results section. We have also included a table in Section 2.6 listing the different products compared in the study, which we hope will clarify the differences between them.*

Specific comments:

1.  l. 34-35 Could you rephrase this sentence for clarity?

    *Revised for clarity as follows: "The latter wavelength is similar in size to the snow grains, which induces significant volume scattering and attenuates the signal (Chang et al., 1987; Kelly et al., 2003; Mätzler, 1994)."*

2.  l. 38 It would be good to give explicit examples of "other SWE products".

    *Following examples added:*

    *" the NASA Global Land Data Assimilation System version 2 – GLDAS-2; the European Centre for Medium-Range Weather Forecasts (ECMWF) interim land surface reanalysis – ERA-Interim/Land and ECMWF Reanalysis version 5 – ERA5 and the Crocus snow model driven by ERA-Interim meteorology"*

3.  l. 44-45 This may be trivial, but I think some readers would appreciate a sentence or two explaining the physics behind this method limitation.

    *Following explanation added to text:*
    *"This occurs because, at higher frequencies (~37 GHz), snowpack transitions from a scattering medium to an emitter when SWE exceeds ~150 mm, reducing sensitivity to further SWE increases."*

4. l. 52 Can you briefly explain why only the March SWE time series has been thoroughly evaluated by this method?

*March is commonly evaluated due to peak snow mass and data availability, as noted in Pulliainen et al. (2020) and Luojus et al. (2021). This rationale is now explicitly stated.*

5. l. 68-92. This should be included in a new subsection of its own.

*Created new subsection titled "2.1 SWE retrieval".*

6. l.70 Can you expand on the source of the snow depth measurements.

*Sources specified and text edited as follows:*
*"SD measurements are collected from multiple sources. The main sources for Eurasia are the European Centre for Medium-Range Weather Forecasts (ECMWF) and the All-Russia Research Institute of Hydrometeorological Information - World Data Cente (RIHMI-WDC) (Bulygina and Razuvaev, 2012). Global Historical Climatology Network daily (GHCNd) (Menne et al., 2012) by National Oceanic and Atmospheric Administration (NOAA) is used as the main dataset for North America."*

7. l. 83. Can you explain why the average of the 6 closest snow depth measurements is calculated and what effect this would have on the variance if the measurement points were sparsely distributed?

*The average of the 6 closest stations is calculated to smooth the results. Using single values creates a "bull's eye effect". Sparsely distributed points often mean a bigger difference between estimations (snow conditions are often more different when the distance between points is larger) which increases the variance of snow grain size estimates. This in turn reduces the weight given to the radiometer data in the final SWE estimate.*

8. l. 90 (and l. 106) How was this constant value of snow density defined?

*The constant of 240 kg m$^{-3}$ is used based on Sturm et al (2010)., added mention of this to text.*

Sturm, M., Taras, B., Liston, G., Derksen, C., Jonas, T., Lea, J.: Estimating snow water equivalent using snow depth data and climate classes. J. Hydrometeorol. 11, 1380–1394, https://doi.org/10.1175/2010JHM1202.1, 2010.

9. l. 111-114. Please state the difference between the new and original threshold detection values.

*Added differences between threshold values to text: "The threshold for SD was decreased from 80 mm to 30 mm, the brightness temperature thresholds were changed from 250 K to 255 K for Tb37V and from 240 K to 250 K for Tb37H."*

10. l. 121. Consider "spatial interpolation and assimilation" as a subsection of the first part of the method (see. Specific comment 5.).

*See answer to comment 5.*

11. l. 130-132. Snow courses may not be evenly distributed throughout the winter neither.

*That's true, temporal distribution is not even. However, Figure 1 shows temporal distribution of reference SWE dataset, and the average date of observation is relatively close to middle of the month.*

[Figure]

*Figure 1 Temporal distribution of reference SWE estimates.*

12. l. 132-133. Is the bias really stable over time? Please explain the implications of this assumption.

*As discussed in Sect. 2.5 (old Sect. 2.4) the monthly bias is relatively stable over the long term. In North America there's no trend in the bias. In Eurasia, there is a small trend in bias over the 40-year period (Appendix A), but it is not statistically significant. We have added P-values to Appendix A to support this statement.*

*A small trend in bias (in Eurasia) is worth examining more closely in future studies. One potential approach could be to evaluate bias in separate time periods (e.g., by decade). If large enough, a trend in bias can act to reduce or reinforce intrinsic trends in long-term SWE estimates (e,g, due to climate change).*

13. l. 171. I suggest to expand on the source of snow depth, SWE and snow density measurements.

*New information about sources of snow depth added, see comment 6. Sources of SWE/snow density measurements are listed in section 2.5 'Summary of changes in the refence snow course data'. For the SWE/snow density data we tried to limit the duplication from other papers and instead reference the publicly available NorSWE dataset (Mortimer and Vionnet, 2025) which contains a detailed description of the data used here.*

Mortimer, C., and Vionnet, V.: Northern Hemisphere in situ snow water equivalent dataset (NorSWE,1979-2021), ESSD Discussions,https://doi.org/10.5194/essd-2024-602, 2025.

14. l. 211. Figure 1:

1. The figure can be enlarged to make it easier for the reader to read and interpret.

2. I am struggling with the black and gray colors on the figure. What are these colors supposed to represent?

*Figure enlarged and legend added to the figure.*

15. l. 227-228: Can you elaborate?

*We use the same in situ data to compute the bias fields for two different SWE products (GSv3.0 and SCv3.1). Differences in the resultant bias fields are therefore a reflection of differences in the products rather than differences in the in situ data used to derive the bias correction fields. The main differences between these two products (retrieval and input data) are the Tb data, the use of spatially and temporally varying snow densities instead of a constant value, and updated snow masks. These changes are outlined in Sect. 2.2. We have added reference to that section in this sentence.*

16. l. 256. Figure 2. Same comment as Figure 1 regarding the colors.

*Legend added to Figure 3 as well.*

17. l. 268. Original = old and updated = new? Please be consistent throughout the manuscript.

*Standardized terminology to "original" and "updated" throughout.*

18. l. 278-279. What would cause this negative bias in spring? Is it related to density assumption? Please elaborate.

*The negative bias during spring is partially due to constant density which tends to be too small during the spring season. This was demonstrated in Mortimer et al. (2022) which compared the difference in observed snow density from the reference snow courses (using an older reference dataset) versus the static value. They showed (Figure 11 in Mortimer et al. 2022) that the static value overestimates until about February and then underestimates. Additionally, wet snow is common during spring which makes retrieval challenging as only the top most layer above water layer is seen by the radiometer.*

19. l. 282; l. 361; Table 1. SCv3.1 instead of SCv3.0?

*Yes, it should be SCv.3.1*

20. l. 293-294. This sentence is redundant. Can you please rephrase it?

*Rephased as follows:*

*In April and May, the magnitude of the negative bias is larger in the updated fields, whereas in March, it is lower.*

21. l. 304. "...*where the snow mass from the updated bias correction (blue line) is > 100 Gt larger than when calculated* **with** *the original reference data (black line).*"

    *Rephrased for clarity as suggested.*

22. l. 307. Figure 5: Do you also see a similar difference between the products for April snow mass?

    *Yes, differences in snow mass between products are similar in April (shown below in Figure 2) and because of this we didn't include figure for April in the manuscript.*

[Figure]

*Figure 2 Average April snow mass*

23. l. 313. Names can be somewhat confusing (see Specific comment 19).

    *Changed "Snow CCI" to "SC" throughout.*

24. l. 327-328. It would be interesting to discuss why the addition of new snow courses increases the negative bias between the two GS data products.

    *The new sites are in regions (e.g. Alaska, Quebec) where SWE can exceed the retrieval algorithm's saturation limit (~150 mm). When this occurs, the bias will be negative because the retrieval algorithm cannot estimate these higher SWE values observed in-situ. Even when the retrieval relies solely on the interpolated SD field (e.g. wet snow), an upper limit is placed on the allowable maximum SD to ensure convergence during assimilation. By adding sites where the SWE often exceeds the method's upper limit we introduce more instances of*

*negative biases. Further, because SWE can be large at these locations the negative biases can be large.*

25. l. 332. Are the differences really most pronounced in May compared to other months? It is not possible to tell from Figure 6 alone.

    *Yes, differences are most pronounced in May. We acknowledge it is a bit challenging to see on Figure 6 because we limit the colormap to +/-100mm to highlight the smaller differences. However, many of the differences are near these +/-100mm values whereas there are much fewer areas with such large differences in the other months shown. This can be seen also in the (new) figure 7 (old figure 8), where differences between monthly bias corrected snow mass estimates are largest in May.*

26. l. 342. Figure 5 instead of Figure 4?

    *Yes, should be Figure 5.*

27. l. 352. Here you refer to figure 8 before figure 7, which makes it confusing. I think that reorganising section 3 (see General comment) would make it easier to refer to figures in the order in which they appear in the text.

    *Figure 7 and 8 have been reordered so they appear in order they are mentioned in text.*

28. l. 371. Table D1 (instead of Table B1).

    *Fixed*

29. l. 412 Figure 7:

    1. For consistency, I suggest using hemispherical maps as shown in the previous figures.
    2. What do the gray areas represent on the maps? Complex topography? Please state this clearly in the caption.
    3. The style of the caption for Figure 7 does not follow the requirements of the journal. Please revise.

    *Added explanation of complex terrain to caption and reformatted caption to follow requirements of the journal.*

30. l. 425-433. Consider adding a supplementary figure showing the locations of SWE > 150 mm and SWE > 200 mm. This would help the reader in interpreting your results.

*Large SWE values are present in different locations depending on the time of year and the year in question, so showing their locations might not help readers that much. Appendix E shows the percentage of large SWE values in different reference datasets.*

31. l. 452. Typo. SWE instead of SEW.

    *Fixed*

32. l. 457-464. It would be interesting to further discuss how the new GSv3.0 and SCv3.1 data products may increase the bias in Quebec and Alaska (which are high snow locations).

    *The products themselves do not increase the bias in Quebec and Alaska. Rather, adding new sites to the reference data used to evaluate the products altered the bias.*

    *The bias calculated with the updated reference dataset more appropriately captures the true bias in both the GSv3.0 and the SCv3.1 products. However, we acknowledge there are still many gaps in the reference dataset so some regions (e.g. arctic tundra) may not be fully capture in our validation. As discussed in response 24, adding sites in high SWE areas adds negative bias because SWE in these areas often exceeds the method's detection limit, when this occurs GS/SC will underestimate SWE compared to what is measured in-situ.*

33. l. 466-468. Is it the opposite in April and May (snow density higher than 240 kg m$^{-3}$). It seems so from the bottom row of Figure 4. Please expand on that.

    *Yes, the constant snow density tends to be too small in April and May which contributes to the underestimation of SWE seen in GSv3.0 product (see response 18). Changing to the spatially and temporally varying snow density (SCv3.1) has reduced the negative bias during these months. Added explanation of this to text.*

34. l. 480. It would be interesting to further discuss the role of spatial resolution and how it affects the bias fields compared to the effect of other factors (such as a constant density assumption).

    *We have added more information about the effects of spatial resolution to the text. More detailed analysis of the change in PMW input data can be found from the Mortimer et al. (2022).*

    *Overall, a finer grid spacing provides some improvements to the correlation, RMSE and bias of the SWE retrieval but has minimal effect on northern hemispheric snow mass (differing from snow density which has significant effect in snow mass).*

35. l. 502-504. As North America is presented before Eurasia throughout the manuscript, I suggest that this order be maintained in the conclusion.

    *Order changed in conclusion.*

---

## Author Comment (AC2)

Reply to Brenton Wilder

*We thank the reviewer for their time and constructive comments on the manuscript. We take all the comments and suggestions into account. Our replies are written in italic font.*

**Larger comments to authors:**

- Daily bias correction using linear (e.g., MLR) and non-linear methods (e.g., random forest) have shown success in previous work estimating SWE bias from SNODAS (King et al., 2020). With your ample amount of snow course data and large study domain, have you considered using such methods to estimate daily bias fields for PMW? The reason I suggest this is because you inherently lose valuable temporal information by first aggregating to monthly fields for the purpose of kriging, correct? If my statement is true, I think a more nuanced discussion may help inform present/future efforts?

  *Although the snow course data cover a large spatial domain, their infrequent sampling, on average every two weeks, does not lend itself to direct calculation of daily bias fields. To have sufficient information to capture the spatial structure of the SWE biases it was necessary to first pool all information at the monthly scale. The daily bias-correction fields are then linearly interpolated from the monthly ones to provide a daily bias-corrected SWE product. We recognize that this approach has limitations, and it does mean we lose some temporal information when we estimate the daily biases.*

  *Further, we acknowledge that there may be more advanced bias correction methods (machine learning) which could be studied in the future. We have added the following paragraph to discussion:*

  *"In this paper simple mean bias subtraction was used as the focus was on updating an existing and proven method and expanding it to new months and to a higher temporal frequency (daily instead of the original monthly). Our results provide a baseline against which bias correction methods could be studied in the future. For example, King et al., 2020 studied different SWE bias correction methods at local scales, finding that random forest techniques provided accurate results, indicating the potential of machine learning-based approaches for bias correction."*

- Another point on the daily bias via monthly bias fields, and correct me if I am wrong, but this hinges on two things: 1) that DMSP SSM/I-SSMIS is a daily product (which is true), and 2) that snow course data is uniformly distributed across a given month, temporally? For example, if more snow course observations were present at the frontend or backend of a given month, there would be issues with interpolating based on the 15th? How might this impact your Eq 1-3?

  *Correct, our method assumes that the in-situ data pooled within a given month provides a robust estimate of the bias. This assumption will depend somewhat on the temporal distribution of snow course data. We have explored two aspects of this assumption below.*

1) *Timing of snow course measurements over the month.*
   *Snow courses are sampled according to a set schedule set out by the operating agency as detailed in Mortimer and Vionnet (2025). This schedule differs by agency but most commonly occurs every 2 weeks, centered around the beginning and middle of the month, or middle and end of the month. Some agencies only sample once per month either at the beginning or end of the month. The sampling frequency in the Russian network increases during the snowmelt period. To better determine the potential effects of such sampling differences on our results, we investigated the temporal distribution of in situ observations, shown in Figure 1. The mean observation date (dotted vertical line) is close to the middle (15th) of the month in February and March but is shifted later in December and January and earlier in April and May.*

[Figure]

Figure 1 Temporal distribution of snow course data.

*The sampling bias seen in Figure 1 during the shoulder seasons occurs because snow courses are only collected when there is snow. Because of this, snow-free conditions common during the accumulation and ablation season are not captured. The monthly mean computed from the reference data will, therefore, tend to overestimate the true monthly mean especially during the shoulder seasons. This will affect the monthly bias correction fields, which inform the daily bias correction. From Figure 1, it is clear that in May, observations are concentrated in the first half of the month. If the absence of observations is due to an absence of snow, the biases might be overestimated. We study this by extrapolating the daily bias correction field during December and May.*

*We extrapolated bias data for the end of May and the beginning of December and generated new bias maps utilizing these extrapolated data. We used simpler linear model to extrapolate bias values for locations for which we had some observations for end of April/beginning of May. Figure 2 shows the difference between this new extrapolated bias field and the original bias field made without extrapolation. Figure 3 shows differences in snow trends corrected using these two bias maps. In December, the differences are small, with extrapolation increasing snow mass estimates by ~30 Gt. In May, the differences are more pronounced, and extrapolation decreases snow mass by ~50 Gt.*

[Figure]

Figure 2. The absolute difference in bias fields made using extrapolated data and without extrapolated data.

[Figure]

Figure 3 December and May average Northern hemispheric snow mass (without mountains) bias corrected using the updated SCv3.1 bias correction fields discussed in the article and using bias correction field that utilize additional extrapolated data.

*2) Apart from the sampling issues during December and May, how sensitive are the monthly and daily bias correction fields to choice of monthly bin edges?*

*To further study how the temporal distribution of refence data affects our results, we tested re-calculating monthly bias fields centered on 1st of each month (instead of the 15th). For example, the "January" bias correct field was made using data pooled from between Jan 15-Feb 15.*

*Then we interpolated daily bias fields from these recalculated "monthly" bias fields (weighted means between the first and last day of each month). For example, daily fields for days in December were interpolated using the monthly maps from "November" (in situ data pooled from Nov 15-Dec 15) and "December" (in situ data pooled from Dec 15-Jan 15). In order to compare these new daily interpolated fields to the old ones, we calculated the monthly average of the daily fields, for example, between Dec1 and 31and compared this to the original monthly bias correction fields (the originally calculated December bias correction field in this case). Figures 4 and 5 below show the difference between these two methods for*

*December, February, and May. In February and December, the effect of changing the center bin is minimal – the blue triangles and circles overlap. In May, we can see that centering the bias field around the 1st of the month increases snow mass little. Overall, the decision of around which data we center our interpolation (e.g., 1st vs. 15th of the month) has an effect on the bias correction. The impact is limited/none during the middle of the snow season but is measurable/larger in May. We could consider limiting the bias-correction to December through April (or January-March/April).*

*We will add discussion about these two topics to the article.*

[Figure]

Figure 4 December, February, and May average Northern hemispheric snow mass (without mountains) bias corrected using the SCv3.1 bias correction fields discussed in the article and using bias correction fields made with different bins,

[Figure]

Figure 5 Difference in bias fields between made using moved bins and original bins.

- Is it common to fill missing values prior to avg monthly bias using this technique? It would seem this could potentially introduce a bias itself? How does this gap filling impact your daily model? You return to this idea in Line 485 for example, where you state the end of May can have positive bias. Do you include June in the interpolation? Also, similarly does filling missing values impact December in a similar way? What is the purpose of filling and not just using the geometric mean? I think I would just like to see clarification in the reviewer discussion and not necessarily in the manuscript.

  *To clarify: neither the monthly nor daily bias correction fields contain gap-filled data. The monthly bias-correction fields are based on pooled in situ data as it is available across the given month. The daily fields are linearly interpolated from the available monthly fields but this interpolation process is distinct from the gap filling. The daily frequency bias-corrected SWE product is only provided on days with available algorithm retrievals, so it does not contain any gap filled data.*

  *The gap filling \*is\* done only when calculating the monthly mean SWE to which the monthly bias correction is applied. Data may be missing either because there is no satellite information for a given day (during the SMMR period satellite information is only available every other day with two gaps of several weeks) or because the SnowCCI algorithm did not produce a retrieval. The latter occurs more frequently during May, but for November through April is rare. For years when we have multiple missing days in May (near the end of the month) monthly mean SWE is biased high if we calculated it using only the available data. When we fill the missing SWE estimates, we get lower, more realistic, monthly estimate.*

  *Line 485 is incorrect. The GSv3.0 May monthly mean is too large and SCv3.1 May estimate is much more reasonable, partially due to filling the missing days and thus reducing the mean value. This will be fixed in the text.*

- Clarification on 40 years of data which you mention in the text. However, you refer to analysis 1980-2018 (39 years). Is this a typo or is it 39 years of analysis?

  *Thank you for noting this inconsistency. The monthly analysis spans 1980–2018 (39 years) as this is the period for which the GSv3.0 product is available, while the daily analysis extends to 2022. We'll revise all relevant sections to ensure clarity and consistency.*

**In-line comments to authors:**

Line 19: I'm not sure necessarily if these products need to be mentioned in the abstract but will leave it up to the authors.

*Removed listing of products.*

Line 35: 37GHz ~ 8 mm. I don't think this statement is entirely true as most snow grains are between ~ 0.5-4 mm? It's certainly closer than 19GHz, but I think I would like to see a response from the authors or to adjust the language just slightly.

*Thanks for pointing this out, we have adjusted the language to be more accurate:*

*"The latter wavelength is closer in scale to the snowpack microstructure, which induces significant volume scattering and attenuates signal (Chang et al., 1987; Kelly et al., 2003; Mätzler, 1994)."*

Line 45: At this specific frequency? Maybe it could be useful to specify. E.g., snow is a scattering medium in visible wavelengths for all values of SWE.

*Added following sentence for clarification:*

*"This occurs because, at higher frequencies (~37 GHz), snowpack transitions from a scattering medium to an emitter when SWE exceeds ~150 mm, reducing sensitivity to further SWE increases."*

Line 49: Listing the 4 products like this is a little awkward, but perhaps could revisit the grammar to include them all together in a list.

*Sentence updated as follows:*

*"Pulliainen et al. (2020) applied this concept to four snow products: MERRA2, GlobSnow GSv3.0, and the Crocus and Brown snow models, both of which were forced by ERA-Interim. This reduced the spread in March SWE estimates from 33 percent to 7.4 percent."*

Line 56: GS has not been defined in the text.

*Added definition to the earlier section.*

Line 61: I suggest the authors consider if this paragraph is needed.

*Removed paragraph.*

Line 75: You mention masking complex terrain from retrieval but discuss results across mountains (e.g., US Western Mountains in Line 326), please explain?

*GlobSnow masks out complex terrain rather than mountain regions which is an important distinction in definition. The reason for exclusion of complex terrain is that the SD data do not capture the high within-grid-cell variability of SD in complex terrain (Takala et al. 2011; Luojus et al. 2021).In the GlobSnow retrieval algorithm a grid cell is considered to be complex terrain if the standard deviation of the elevation within a grid cell is more than 143 m. The complex terrain mask does not exclude all mountain areas. Many high elevation mountain plateaus are retained. These areas tend to have SWE in excess of the method's detection limit which can lead to large errors. The presented bias correction helps address such biases.*

Luojus, K., Pulliainen, J., Takala, M., Lemmetyinen, J., Mortimer, C., Derksen, C., Mudryk, L., Moisander, M., Hiltunen, M., Smolander, T., Ikonen, J., Cohen, J., Salminen, M., Norberg, J., Veijola, K., Venäläinen, P.: GlobSnow v3.0 Northern Hemisphere snow water equivalent dataset, Sci Data, 8, https://doi.org/10.1038/s41597-021-00939-2, 2021.

Takala, M., Luojus, K., Pulliainen, J., Derksen, C., Lemmetyinen, J., Kärnä, J.P., Koskinen, J., Bojkov, B.: Estimating northern hemisphere snow water equivalent for climate research through assimilation of space-borne radiometer data and ground-based measurements, Remote Sens Environ 115, 3517–3529, https://doi.org/10.1016/j.rse.2011.08.014, 2011.

Line 76: What method is used for wet snow to relate SD to SWE when the PMW algorithm is not used?

*When wet snow is detected, SWE is estimated from the interpolated SD field. In GSv3.0, we use a constant snow density value of 240 kg/m3 to go from SD to SWE. In Scv3.1 we use spatially and temporally varying snow density fields as described in* Venäläinen *et al. (2023). These fields are based on the same snow course data used in the present bias corrected.*

Line 79: There is often ambiguity and both are commonly used, please specify if you are referring to diameter or radius for effective snow grain size. I assume diameter based on d0 but would just like to clarify.

*Yes, we are referring to diameter. Added clarification of this to text.*

Line 84: Should use previously defined d0 symbol.

*Change to $d_0$ symbol.*

Line 90: You mention later on that constant density = 240, I suggest adding here as well for the reader.

*Value added.*

Line 94: Is it worth having the CDR acronym as it only is stated once in the paper?

*Removed acronym.*

Line 150: What exactly do you mean when you state, "filled from two closest"? Linear interpolation? If so, it would be nice to state this.

*The mean of the two closest observations is used, clarified this in text.*

Line 175-176: Suggest revising to simpler sentence structure here.

*Updated sentence as follows:*

*"Although both the density fields and bias correction use snow course data, they include different observations and employ slightly different data aggregation methods."*

Line 194: Appendix A figure may be more supportive to your claim if you were to include a simple statistical trend test to assess for slope, significance levels, etc based on the data points shown here.

*Added p-values to Appendix A.*

Line 208: The updated SWE being larger is not visually apparent in the histograms, and could be aided by adding the colored mean values (red/blue) as text to the plot, and/or adding dashed vertical lines representing the mean SWE.

*Vertical lines representing monthly mean SWE values have been added to figure.*

Figure 1&2: Please be consistent with "old" vs. "new" / "updated" vs. "original". You should stick with only one of these and stay consistent throughout the text to be more concise.

*Standardized terminology to "original" and "updated" throughout.*

Figure 1&2: What is being shown with the black/white and the green backgrounds?

*They show the monthly average mean SWE, color bar added to figure and explanation to caption.*

Line 235-239: Revise run-on sentence.

*We split the run-on sentence into two sentences as follows:*

*"However, averaging and interpolation steps are applied to these data and automated data are included to compute the density fields. This means that the individual in-situ samples are not fully correlated with the bias-corrected (or non-bias-corrected) SCv3.1 estimates, nor are they fully independent."*

Line 246: You can add in this sentence, "because of gammas higher relative accuracy", or something similar to this effect.

*We chose to refrain from adding qualifiers about the relative accuracy of the various reference datasets. Constraining the accuracy of in situ SWE method and observations is challenging because it varies according to observer and environmental factors (see Mortimer and Vionnet (2025) and references therein). The difference in spatial scale of the analysis versus the measurement footprint is an additional consideration. Direct comparison of airborne gamma SWE and snow course measurements showed good agreement between these methods in non-mountainous areas of eastern North America up to spatial scales of at least 50km (Mortimer et al. 2024).*

*The key point about the airborne gamma SWE is that they are completely independent of any data used to derive either the bias correction or the density fields. Because the airborne gamma method does not provide separate density estimates so they cannot be used to derive the spatially and temporally varying density fields applied in SCv3.1. Further, the airborne gamma observations are typically conducted only once per year, so we did not include them in the SWE bias correction fields. We included more information about the temporal distribution of the airborne gamma data in Sect. 2.6.*

"*Airborne gamma observations are typically conducted once per year near peak SWE. Less than one third of site-years have more than one observation. Observations are concentrated in February and March (32% and 38% of observations, respectively). Observations in December, and May each account for less than 1% of the data.*"

Line 250: Not required to add, but may be insightful to add here for early-career readers to very briefly (~1 sentence) discuss "saturation effect" (see example in Cho et al., 2020) and the rationale for splitting validation metrics at 150 mm SWE.

*We have added explanation of limited ability to observe large SWE values to the introduction.*

Line 256: This is the first mention of SnowPEx. Briefly introduce, as well as discuss why you are comparing this to your NH SWE estimates. (I understand after reading it for the first time, but extra background here I believe will strengthen your methods).

*We have updated the Figure to only include the shading from SnowPEx+ and referenced the range of products as those found in Mudryk et al. 2025.*

Figure 3: Is it not clear to me if the black/white areas are different snow climates? Extents of the model?

*Updated figure to include color bar with explanations of different colors.*

Figure 5: Shouldn't you also mention this is excluding complex terrain as done in Figure 8? Capitalize the Northern Hemisphere (check throughout text)?

*Good point. We now mention that complex terrain was excluded and have capitalized Northern Hemisphere.*

Line 295: With the over-abundance of SD for informing inversion in Finland (and stable bias), I am curious if future work could try holding out SD/SWE for independent evaluation (if data available).

*Yes, in previous studies (*Venäläinen et al., 2021; 2023) *we divided SWE data into development and validation locations. To ensure as extensive a dataset as possible for the bias correction we chose not to hold out any data for validation in this study.  In the future we will consider holding out some data from areas with spatially dense networks (e.g. Finland) for validation.*

Venäläinen, P., Luojus, K., Lemmetyinen, J., Pulliainen, J., Moisander, M., and Takala, M.: Impact of dynamic snow density on GlobSnow snow water equivalent retrieval accuracy, The Cryosphere, 15, 2969–2981, https://doi.org/10.5194/tc-15-2969-2021, 2021.

Venäläinen P, Luojus K, Mortimer C, Lemmetyinen J, Pulliainen J, Takala M, Moisander M, Zschenderlein L.: Implementing spatially and temporally varying snow densities into the GlobSnow snow water equivalent retrieval. The Cryosphere, 17(2), https://doi.org/10.5194/tc-15-2969-202,2023.

Line 301: Non-mountainous correct?

*Yes, added mention of this to text.*

Line 331: I was thinking a lot about this sentence, and have a suggested rewrite the authors may consider:

"Applying the updated snow course data to both GSv3.0 and SCv3.1, we show ..." etc.

Something like this? The point is this is getting at the retrieval algorithm and so it may be helpful to lead with the updated snow course data being the same in this scenario.

*Sentence updated.*

Line 337: It would be best not to say probably, and if you have these data you should be able to verify.

*We cross-referenced these areas with snow course sites which confirmed that these localized differences correspond to snow course locations. We have removed mention of the synop data from the sentence.*

*Additionally, comparisons using the un-bias-corrected monthly products show much smoother results, indicating that the synoptic data set matches quite weakly over the 39-year period.*

Line 345: Be more specific, are these changes to the retrieval related to the variable snow density?

*Yes, these changes in March snow mass in Eurasia are related to changes in the retrieval algorithm and input data as outlined in Section 2.2. We have added reference to Section 2.2 to this text.*

Line 348: Please add the years discussed in Mortimer et al 2022 in this sentence.

*Added mention of years to sentence.*

Line 351: Instead of saying, "...to that of two suites of reanalysis products, as described in Sect. 2.5..." Please consider simply referring to SnowPEx here.

*Change to reference to SnowPEx.*

Line 370: Referring to section 2.5 like this is confusing since it also includes gamma data. Can you not just state SnowPEx reanalysis data here?

*Added mention of SnowPEx to the sentence for clarity.*

Line 371: I think you mean Table D1**. Also, I am curious if you tested the daily bias correction SCv3.1 and the monthly bias correction SCv3.1 against gamma. It appears in this table you have just done the daily? However, would it not be beneficial to show if there is improvement from your new approach over the prior method?

*We only validated daily data against gamma SWE measurements. There were insufficient airborne gamma data within a month for meaningful validation of monthly bias corrections .*

Table 1: Why isn't V3.1 included in Table 1 against gamma SWE measurements? Or is this a typo?

*Yes, it's a typo, fixed.*

Line 394:  I know the reanalysis product is not a pure validation set, but I think it still may be helpful to see Pearson correlation for Figure 7. I'm thinking this decision is best left to the authors however.

*We tested whether the bias correction affects the pattern correlation between the reanalysis and SC3.1 datasets. The results indicate that the bias correction slightly reduces the pattern correlation with the reanalysis, although the effect is minor. This small degradation is most pronounced over North America, where the magnitude of the bias correction is also largest. Given the limited impact of this effect, we do not consider it necessary to include these results in the article.*

[Figure]

Figure 6. Pattern correlation between SCv3.1 and reanalysis products and between bias-corrected Scv3.1 and reanalysis products.

Figure 7: Is the light grey in this map the topography mask? If so, this needs to be stated in the caption.

*Yes it is, added explanation to caption.*

Figure 8: It seems the months are not aligning with the x-axis ticks (it aligns with the text)? Also there are three shaded areas? In the text you mention only the SnowPEx being shaded?

Figure 8: It would be helpful to draw the mean/median SnowPEx as you refer "to center" often, however, it would be much more clear with it drawn on. Further, you may discuss the mean/median values and the differences between the SCv3.1.

*Figure 8 (new figure 7) will be updated. Only one shaded area is included and model mean values will be added to the figure.*

Line 440: And perhaps areas for future testing could be with region specific reanalysis data, such as those presented in Fang et al. (2022)?

*Yes, local analyses using regional products (e.g. Fang et al., 2022) could help to better understand the SWE retrieval in problematic areas that are identified in our hemispheric scale analysis. Targeted region-specific improvements based on such analysis might yield better local SWE estimates. However, the purpose of the GlobSnow retrieval is hemispheric scale SWE estimates and there is always a tradeoff between local versus global improvements. Added text:*

"*In the future, comparison with region specific reanalysis data might help to better understand the local behaviour in these regions which may help guide future algorithm improvements.*"

Line 447: But is the peak really indicative here? Wouldn't the percentage of observations over, say 150mm, be more physically meaningful? Similar to how you described the 3x increase in low-bin for the updated snow course data.

*We agree that peak SWE may not be the most meaningful metric here and have removed the text attributing differences directly to peak SWE. We have added additional text to better articulate some of the factors contributing to differences in assessed accuracies between airborne gamma and snow courses. The first point is that the validation statics based on the airborne gamma data are not fully representative of the Northern Hemispheric performance because of the limited spatial coverage of the airborne gamma data. Second is the independence of the airborne gamma data and how that may affect the assessed accuracies.*

"*Given the dependence of the bias correction on the snow course data, it is not surprising that validation statistics obtained using those data outperform those based on the airborne gamma data. For example, central North America is well covered by airborne gamma but not by snow courses which are used to develop the bias correction. Consequently, the larger errors obtained when assessed with airborne gamma partly reflect the inability of the bias correction to correct biases in areas with limited in situ information. This highlights the limitations of the bias correction in regions with sparse or no in situ data. Unfortunately, since the airborne gamma data do not cover all snow classes or the full winter season, we are unable to discern whether the magnitude of the errors obtained with airborne gamma apply to other regions*

*Additionally, the difference in the timing and SWE distribution of the two validation datasets may also contribute to the differing accuracies when calculated using snow course and airborne gamma.*

*Previous work (Mortimer et al. 2022 Figure 6) has shown that errors in the SCv3.1 product increase over the course of the snow season in concert with SWE magnitude. Airborne gamma surveys are usually conducted once or twice per year and are concentrated in February and March near peak SWE (Sect. 2.6). Snow courses are conducted more frequently (~ 14 to 30 days in North America (see Mortimer et al. 2024)) and tend to cover the full snow season. This means that snow courses are more likely to capture lower the SWE values of the accumulation season which are often associated with smaller absolute errors (see for example, Mortimer et al. 2020). In contrast, the airborne gamma observations are biased towards higher values during the middle and end of the snow season which are both associated with larger absolute errors. To demonstrate, Figure E1 shows that the SWE distribution of the airborne gamma is shifted higher compared to snow courses, although the snow course data cover a much larger range.*"

Line 455: If this is a question of differences in magnitude, there are validation metrics that account for this such as "relative root mean squared difference" or using percent differences? Perhaps I am looking at this incorrectly.

*Additional statistics might be useful. However, we feel the biggest challenge with the airborne gamma dataset is the limited number of large observations it has as bias correction is most effective for large SWE values. We have revised this section as outlined in our response to L447 to highlight this.*

Line 490-491: I would avoid saying "might" here. If possible, would you be able to check the data and report with added certainty here?

*Figure 7 below shows scatter plots for bias corrected and un-corrected daily products against gamma ray validation data. Bias correction has created number of outlier values by over correcting small SWE values. Removed "might" from the text.*

[Figure]

Figure 7. Scatter plots of bias corrected and un-corrected SnowCCI v3.1 products against gamma ray validation data for SWE < 500 mm.

Line 506: Closer to the mean? Median? How much more accurate was your daily bias product vs. your improved monthly product. To reiterate on one of my points above, if you compared both the monthly and daily bias correction methods to the gamma dataset, you could get at a regional percent improvement, correct?

*The Northern Hemisphere snow mass of the uncorrected daily SCv3.1 product is outside of the range (below the ensemble spread) of the current suite of reanalysis products (SnowPex+). After applying the bias correction, the NH snow mass increases such that it falls within this range.*

*Your suggestion to compare the percent improvement obtained by going from the monthly to daily product is valid and would be a nice way to demonstrate the benefit of the daily (versus monthly) product. Unfortunately, we feel there's insufficient airborne gamma measurements (insufficient temporal frequency) to effectively assess the monthly bias corrected product. There is usually only one airborne gamma SWE measurement per site per year. This instantaneous value does not necessarily represent the monthly mean.*

*All the typos, mistakes and missing info listed below have been fixed:*

Line 10: Suggest changing "global snowpack" instead over "snow cover"?

Line 12: Define (PMW) here for the abstract.

Line 25: GS has not been defined in the abstract.

Line 72: I think you are missing GHz after 19.40?

Line 151: typo, "Filling"

Line 174: Check grammar, "are" used?

Line 180: Check grammar, "'are" not used'?

Line 226: SCI? Do you mean SC?

Line 244: Please provide reference to gamma SWE dataset if available.

Line 269: SC.v31 typo.

Line 280: In the caption, "SCv3.0" should be changed.

Line 313: I suggest to be consistent and either use SCv3.1 or Snow CCI.

Figure 6: I think this may be a typo and should be "SCv3.1"?

Line 359: Please check the typo, "SnowPEx".

Line 384: Table D1*

Line 429: check consistency throughout , "in-situ" vs. "in situ". Also in L236 and  L241.

Line 452: Typo, SWE.

Line 478: Please add the word "SWE" after GSv3.0.

Line 505: "Northern Hemisphere",  capitalize, check throughout.

Line 511: Please consider adding "in relation to GSv3.0" after "for April and May".

Line 572: Correct typo in reference Brown et al. 2018.

Line 632: A note to the authors that this work was published, here is the citation:

Mortimer, C., Mudryk, L., Cho, E., Derksen, C., Brady, M., and Vuyovich, C.: Use of multiple reference data sources to cross-validate gridded snow water equivalent products over North America, The Cryosphere, 18, 5619–5639, https://doi.org/10.5194/tc-18-5619-2024, 2024.